# Future projections of High Atlas snowpack and runoff under climate change

Alexandre Tuel[1,4], Nabil El Moçayd[2], Moulay Driss Hasnaoui[3], and Elfatih A. B. Eltahir[1]

[1]Ralph M. Parsons Laboratory, Massachusetts Institute of Technology, 15 Vassar St., Cambridge 02139 USA
[2]International Water Research Institute, University Mohammed VI Polytechnique, Lot 660 Hay Moulay Rachid Benguérir 43150, Morocco
[3]Ministry of Equipement, Transport, Logistics and Water, Department of Water, Morocco
[4]Current affiliation: Institute of Geography, Oeschger Centre for Climate Change Research, University of Bern, Bern

**Correspondence:** Alexandre Tuel (alexandre.tuel@giub.unibe.ch)

**Abstract.** The High Atlas, culminating at more than 4000 m, is the water tower of Morocco. While plains receive less than 400 mm of precipitation in an average year, the mountains can get twice as much, often in the form of snow between November and March. Snowmelt thus accounts for a large fraction of the river discharge in the region, particularly during spring. In parallel, future climate change projections point towards a significant decline in precipitation and enhanced warming of temperature for the area. Here, we build on previous research results on snow and climate modeling in the High Atlas to make detailed projections of snowpack and river flow response to climate change in this region. We develop end-of-century snowpack projections using a distributed energy balance snow model based on SNOW-17 and high-resolution climate simulations over Morocco with the MIT Regional Climate model (MRCM) under a mitigation (RCP4.5) and business-as-usual (RCP8.5) scenarios. Snowpack water content is projected to decline by up to 60% under RCP4.5 and 80% under RCP8.5 as a consequence of strong warming and drying in the region. We also implement a panel regression framework to relate runoff ratios to regional meteorological conditions in seven small sub-catchments in the High Atlas. Relative humidity and the fraction of solid-to-total precipitation are found to explain about 30% of the inter-annual variability in runoff ratios. Due to projected future atmospheric drying and the associated decline in snow-to-precipitation ratio, a 5-30% decrease in runoff ratios and 10-60% decrease in precipitation are expected to lead to severe (20-70%) declines in river discharge. Our results have important implications for water resources planning and sustainability of agriculture in this already water-stressed region.

## 1  Introduction

The High Atlas is the major source of freshwater for the semi-arid plains of central Morocco. Much of the discharge of the Oum-Er-Rbia and Tensift, the two main rivers of central Morocco, comes from the mountainous terrain where they begin their course. In this region, precipitation essentially falls at elevations above 1000 m (Boudhar et al., 2009); below that, it is scarce and evaporation is extremely high, leading to minimal runoff. Though located in a rather warm region, the High Atlas rises up to more than 4000 m and often experiences below-freezing conditions between November and March (Boudhar et al., 2009). Consequently, snow is a major component of the regional water cycle (Marchane et al., 2015; Tuel et al., 2020a). It accounts

for a substantial fraction of annual runoff, up to 50% in some mountain catchments (Boudhar et al., 2009), and for most of the runoff during spring, as the wet season comes to an end. Snow cover in the High Atlas is characterized by large inter-annual variability (Marchane et al., 2015; Tuel et al., 2020a), mostly following that in wet-season precipitation, itself largely shaped by the North Atlantic Oscillation (Knippertz et al., 2003; Boudhar et al., 2009).

However, the High Atlas snowpack may be particularly vulnerable to climate change. Climate projections over Morocco – and generally the Mediterranean – agree on robust warming and drying trends under greenhouse gas forcing (Cramer et al., 2018; Lionello and Scarascia, 2018; Drobinski et al., 2020; Tuel and Eltahir, 2020). By the end of this century, average winter temperatures in the High Atlas could be 2-4°C higher, and precipitation 25-60% lower, depending on the emissions scenario (Ayt Ougougdal et al., 2020; Driouech et al., 2020; Tuel et al., 2020b). These combined warming and drying trends will unavoidably lead to a snowpack decline. Yet, few studies have analyzed climate change impacts on the local snowpack and regional water availability. López-Moreno et al. (2017) applied a complex physically-based snow model to observed meteorological data at one station in the Moroccan High Atlas, fitted with observed snow depth at the same location. They found that High Atlas snowpack was less sensitive to warming and drying than that in other Mediterranean-climate regions (10-15% snow water equivalent decline per degree of warming), because of colder snowpack temperatures associated with high latent heat losses. Still, their results pointed to a decrease in average snow duration of 25-30% and in mean Snow Water Equivalent (SWE) of 30-55% by 2050.

Future trends in runoff in the High Atlas under climate change have also been investigated, notably by Jaw et al. (2015) who analyzed simulations with the Variable Infiltration Capacity model forced by regional climate model output. They found a general tendency to reductions in streamflow, with a strong sensitivity to the forcing model's precipitation trends. Tramblay et al. (2018) took a simple water balance approach, equating long-term net precipitation with water availability, to estimate future changes in dam storage across North Africa. In the High Atlas, they projected a 40-50% decline in water availability under business-as-usual by the end of the 21st century. Only one study tried to quantify the impact of climate change on High Atlas runoff by taking snow dynamics into account: Marchane et al. (2017) developed runoff projections for the Rheyara catchment, south of Marrakech and part of the Tensift watershed, by running conceptual monthly water-balance models incorporating a simple parametric snow module. They projected a 19-63% decline in surface runoff by the middle of the century, dependent on model and scenario. Coupled with population growth, such trends, if realized, will inevitably translate into growing unmet water demand, as shown by Ayt Ougougdal et al. (2020) for the Ourika watershed in the High Atlas.

Thus, while it is clear that the region is headed towards a pronounced decline in snowpack and runoff, much remains to be done to quantify that decline at the catchment level and reduce uncertainties. In this study, we hypothesize that a warmer and drier climate will have substantial impacts on the High Atlas snowpack. We develop detailed snowpack projections for the High Atlas under climate change and assess the implications of a declining snowpack on regional runoff. We focus on the Oum-Er-Rbia watershed, a major catchment of the High Atlas. To that end, we apply the methodology of Tuel et al. (2020a) (hereafter T20a), who modeled High Atlas snowpack by applying a simple distributed snow model forced with assimilated remotely-sensed and dynamically-downscaled data. Using satellite-observed snow cover as a baseline for the current climate, we fit and run the snow model with output data from high-resolution regional climate simulations over Morocco obtained by

Tuel et al. (2020b). We then quantify the sensitivity of runoff in seven mountain catchments within the Oum-Er-Rbia watershed to large-scale meteorological and snowpack conditions, and use the results to assess the impact of warming, drying and snowpack disappearance on runoff.

## 2   Study area

With a length of 550 km, the Oum-Er-Rbia is Morocco's second longest river (Fig. 1-a). There are about 4 km$^3$ of available renewable water resources in its basin each year, most of which comes from surface runoff (3.5 km$^3$), the rest being groundwater (Agence du Bassin Hydraulique de l'Oum-Er-Rbia, 2012). 90% of this water is used to irrigate 350000 hectares of fields, accounting for 30% of Morocco's irrigated land, with the rest supporting the needs of two major cities, Casablanca and Marrakech, and industrial phosphate mining. In addition, mountain runoff is used to generate hydroelectricity. The Oum-Er-Rbia river begins its course in the northeastern portion of the High Atlas, near the city of Khenifra; as it flows westwards towards the Atlantic Ocean, it receives major contributions from northward-flowing tributaries originating in the High Atlas (Fig. 1-a). Beyond that, the river continues its course in semi-arid plains which bring little additional runoff.

The climate of the area is rather continental, characterized by a large amplitude in the annual cycle of temperature (Knippertz et al., 2003). Minimum temperatures occur in January, when they range from mild ($\approx$12°C) below 1000 m to cold (-5°C) above 3000 m. Temperatures reach their peak in July, at about 35°C below 1000 m and 10-15°C above 3000 m (Ouatiki et al., 2017). Annual precipitation in the whole basin averages about 400 mm, with a low of 250 mm in the lowland plains, and a high of 800 mm in the mountains to the south (Ouatiki et al., 2017). Most of that precipitation occurs between October and May, when the region is under the influence of North Atlantic westerlies (Knippertz et al., 2003; Tuel and Eltahir, 2018). Mountainous areas also experience substantial precipitation during summer, due to small-scale convection (Born et al., 2008). As a consequence, vegetation outside the lower-elevation valleys is sparse, essentially limited to bare soil, grass and occasional shrubs (Baba et al., 2019). Snowfall is common between November and March above 1500 m elevation, and it is frequent to observe snow cover persisting for several months above 2500 m (Marchane et al., 2015). Inter-annual variability is substantial, however, following that of precipitation (Boudhar et al., 2010). Melt begins in February, and the snowpack is typically gone by the end of May (Tuel et al., 2020a).

We focus specifically on the 13,610 km$^2$ domain analyzed by T20a (Fig. 1-a), which encompasses most of the area above 1000 m altitude within the Oum-Er-Rbia watershed and thus most regions that receive significant snowfall (Marchane et al., 2015). Elevation in this domain ranges from 621 m to 3890 m. with an average of 1882m.

# 3 Data and methods

## 3.1 Data

### 3.1.1 Hydroclimatological data

We use for this study a mixture of model-, station- and satellite-based hydrometeorological data (see Table 1). Model-based data is described in section 3.1.2. Daily precipitation data are available at seven stations in the study area, including three at more than 1200 m elevation, over the 1980-2015 period (Fig. 1-a) (see also Table A1). For each precipitation series, we conduct basic quality checks following Durre et al. (2010). We then discard all the months for which more than 10% of the data is missing or flagged. This leaves more than 95% of the data for analysis. This data is used to check the validity of the satellite precipitation products described below. In addition, daily discharge measurements are available at seven locations as well, between 1978 and 2015. Each has at most 0.5% of missing data. We implement a simple quality control following Gudmundsson et al. (2018). Days with negative discharge are flagged, as well as all consecutive periods of more than 10 days during which discharge values are equal and larger than zero. We also flag as suspect daily discharge values $Q$ such that $\log(Q+0.01)$ is more than six standard deviations away from its mean value, with mean and standard deviation computed over a 10-day period around the corresponding calendar day over the whole time series.

The locations of the discharge stations define seven sub-catchments for which runoff will be modeled, from north to south: Tarhat, Chacha, Ouchene, Tillouguite, Moulay Hassan, Segmine and Tamesmate (Fig. 1-a). Area and average elevation for each catchment are given in Table 2. Their average elevation varies from 1460 to 2360 m. We remove the contribution from base flow by subtracting the minimum monthly discharge value for each catchment and each hydrological year (September-August). This correction is minor for all catchments except Tarhat, the northernmost one, which includes the headwaters of the Oum-Er-Rbia river, and receives a substantial contribution of base flow to its annual discharge. In particular, the flow at Tarhat remains high during summer ($\approx$35% of its wet-season peak), likely due to groundwater discharge from deep mountain aquifers. Annual cycles of corrected monthly discharge are shown on Fig. 2-a.

Satellite-based data is used for basin-wide precipitation, temperature and snow cover. 3-hourly precipitation from the TRMM TMPA (TRMM Multi-Satellite Precipitation Analysis) 3B42 version 7 dataset is used as the reference precipitation dataset for the region. It consists in remotely-sensed data corrected with rain gauge data on a monthly basis (Huffman et al., 2007). The data cover the period 1998 to present, at a resolution of 0.25°. While satellite-based precipitation data suffers from numerous biases (Milewski et al., 2015; Derin et al., 2016; Hashemi et al., 2017), it is often the only option available in complex terrain where stations are scarce. Milewski et al. (2015) and Ouatiki et al. (2017) assessed the accuracy of the TRMM 3B42 V7 dataset in the Oum-Er-Rbia and found that, although unreliable at the daily timescale, it offered satisfactory estimates of precipitation if averaged in space or time. Annual cycles of TRMM precipitation for the seven catchments are shown on Fig.1-b.

For comparison, we also consider the CHIRPS dataset, available from 1981 onwards at a 0.05° resolution (Funk et al., 2015). CHIRPS is produced by combining high-resolution satellite-based precipitation with station and fine-scale topography data. In our region of focus, TRMM and CHIRPS show some differences (Fig. 3-a,b): CHIRPS is notably wetter, particularly near

the Tizi N'isly station. A comparison of monthly values with the four available stations above 1000 m suggests a rather dry bias in TRMM and inconsistent biases in CHIRPS (Fig. 3-c,d). Both datasets have a dry bias over the north-eastern corner

of our domain (around the Kenifra station). Across the four stations shown on Fig. 3, absolute biases range from 13-26 mm for TRMM and from 18-25 mm for CHIRPS. Our purpose here is not to perform an in-depth comparison of the performance of the two datasets, but to test the robustness of the runoff projection results to variability in reference precipitation. To bias-correct regional climate model output for snow modeling (Section 3.1), however, we follow T20a and consider only the TRMM dataset.

Reference surface air temperature is derived from MODIS Land Surface Temperature (LST) product MOD11A1 L3 version 6 at 1 km resolution (Wan, Z., Hook, S., Hulley, 2015). We refer to T20a for details on data filling and correction. Observed snow cover area for the region is extracted from the MODIS Terra snow cover daily L3 product (MOD10A1) at 500 m resolution (Hall and Riggs, 2016). Snow cover is detected using values of the Normalized-Difference Snow Index, based on reflectances in the visible/near infrared and middle infrared. We apply the correction methodology described in Marchane et al. (2015)

which allows to substantially reduce the number of missing data points (which are mainly due to the presence of clouds), and average the data at the weekly timescale as in T20a. We refer to these two studies for a discussion of the accuracy of the MODIS dataset in this region. All MODIS data is available from February 2000 to present. Elevation data is taken from the Shuttle Radar Topography Mission 90-meter resolution dataset version 4.1 (STRM90) (Jarvis et al., 2008), and interpolated to the approximately 1 km resolution of the MODIS land surface temperature data.

**3.1.2 Regional climate simulations**

We use the regional climate downscaling data and projections developed by Tuel et al. (2020b) for the Western Mediterranean, at a 12 km resolution, using the MIT Regional Climate Model (MRCM). MRCM is based on the Abdus Salam International Centre for Theoretical Physics Regional Climate Model Version 3 (RegCM3) (Pal et al., 2007), but with significant enhancements of model physics, and notably a coupling with the Integrated BIosphere Simulator land surface scheme (IBIS). Dy-

140 namical downscaling is performed for ERA-Interim (1982-2011) (Dee et al., 2011) as well as three carefully-selected GCMs from the Coupled Model Inter-comparison Project Phase 5 (CMIP5) (Taylor et al., 2012): MPI-ESM-MR, GFDL-ESM2M and IPSL-CM5A-LR, for the historical (1976-2005) and RCP4.5 and 8.5 (2071-2100) scenarios. Details of the simulations, including model setup and performance, can be found in Tuel et al. (2020b).

**3.2 Methods**

The methodological framework adopted in this study is summarized in Fig. 4. We start from GCM simulations from the CMIP5 archive, dynamically-downscaled with MRCM. We apply bias-correction to the MRCM output, using observed and ERA-Interim data downscaled with MRCM as target, which we then feed into a 1 km resolution snow model over our study region to reconstruct snowpack under the various emissions scenarios. Finally, a statistical model is developed for catchment runoff coefficients (RCs), in order to make projections of runoff under future climate conditions.

### 3.2.1 Bias-correction and statistical downscaling

6-hourly wind speed, specific humidity, air temperature, precipitation, and downward longwave and shortwave are extracted from the MRCM output over our domain, at the 12km MRCM resolution. Because the snow model (section 3.2.2) is run at a resolution of 1km, the data must be downscaled to that resolution. In addition, MRCM output must be bias-corrected, when reliable observations are available. Therefore our approach involves a mixture of bias-correction and statistical downscaling of the data. Air temperature and precipitation in all MRCM runs (ERA-Interim and GCM-driven simulations) are bias-corrected at the 6-hourly timescale using MODIS LST-derived air temperature and TRMM precipitation at their native resolutions as respective targets, via the CDF-transform method (Michelangeli et al., 2009). Alone among the three GCMs, the IPSL-CM5A-LR model exhibits a negative bias in wet days that we correct at each grid cell by randomly generating wet days of magnitude drawn from the corresponding distribution of wet-day precipitation in the TRMM dataset. For bias correction, reference periods for "perfect" observations are 1998-2011 for TRMM and 2000-2011 for MODIS. The corresponding periods in the simulations are the same for ERA/MRCM, and the 1992-2005 and 1994-2005 periods, respectively, for each of the GCM-driven simulations. All bias corrections are performed for the cold (November-April) and warm (May-October) seasons separately. As to other variables, for which reliable observations are not available, the output from the ERA-Interim run (hereafter referred to as ERA/MRCM) is taken as "ground truth". Therefore, wind speed, downward long- and shortwave radiation and specific humidity in the GCM runs is bias-corrected against the ERA/MRCM values over the 1982-2005 reference period.

The downscaling to the MODIS 1km resolution is then obtained as follows. Precipitation, downward long- and shortwave and wind are simply downscaled with no elevation correction, keeping the 12km-grid value for all the 1km grid cells. Specific humidity, however, is downscaled with an empirical lapse-rate $\mu$ estimated at each time step:

$$\log(q) = \log(q_{12}) + \mu \cdot (z - z_{12}) \tag{1}$$

where $q_{12}$ is the specific humidity in a given 12 km resolution grid cell of elevation $z_{12}$, and q the downscaled value at elevation z.

### 3.2.2 Snow model

We apply the SNOW-17 model (Anderson, 2006) with a radiation-derived temperature index for melt (Follum et al., 2015) as described in T20a. SNOW-17 simulates snow accumulation and loss based on meteorological variables, and accounts for the various energy balance equation terms. Snowpack is characterized by its snow water content (SWE) and heat deficit, defined as the amount of heat (in equivalent mm of SWE) required to bring its temperature up to freezing point. We also integrate the bulk-aerodynamic formulation of sublimation detailed in T20a. Readers are referred to Follum et al. (2015) and Anderson (2006) for more details about SNOW-17.

We run the model at a 6-hourly time step and at the native MODIS LST 1 km resolution over the domain depicted in Figure

1-a. Model output includes SWE, as well as melt and sublimation fluxes at each time step. SWE is then translated into snow cover fraction for each grid cell using the following relationship:

$$SC = 0.85 \times \tanh\left(k \cdot SWE\right) \tag{2}$$

with $k = 100$. The snow model requires optimizing three parameters: $M_f$ (melt factor), $NMF_{max}$ (maximum negative melt factor) and $TIPM$ (coefficient used in updating snowpack temperature) (Anderson, 2006). In keeping with T20a, parameter

calibration is performed by maximizing the Nash-Sutcliffe coefficient (Nash and Sutcliffe, 1970) between the annual cycles of observed (MODIS) and simulated snow cover at 250 randomly selected grid points within the snow domain. We force the elevation distribution of these 250 points to match that of the whole domain. The annual cycles are computed for the 1995-2005 period in the GCM-driven simulations, 2000-2011 period in the ERA-Interim simulation and 2000-2011 period in the MODIS series. Parameter are calibrated independently for each of the simulations (ERA-Interim and three GCMs) in their respective

reference periods. For the future simulations, parameter values are kept constant, equal to their calibrated values.

In the analysis of model results, a special focus is given to current and future sublimation fluxes. Due to the particularly arid climate of the High Atlas, sublimation losses are indeed quite significant in our study area: about 9% of all snowfall on average, and up to 30% above 3500 m (Schulz and de Jong (2004); López-Moreno et al. (2017),T20a).

### 3.2.3 Statistical modeling of runoff coefficients

We model catchment runoff coefficients (RCs), defined as total October-May discharge divided by total October-May precipitation, across time and space as functions of large-scale hydrological variables by adopting a panel regression framework. Panel regression allows to enhance the effective size of a dataset and to obtain more robust estimates of the response to selected covariates compared to more traditional regression approaches (Steinschneider et al., 2013; Davenport et al., 2020). It also allows to account for static (space-dependent) and time-varying (time-dependent) factors, although here, with only seven

catchments, we do not have enough data to make robust statements about static factors responsible for the disparity in average RC (Fig. 2-b). Therefore, we focus on time-varying covariates, and consider a fixed-effects formulation:

$$\log\left(RC_{j,t}\right) = \log\left(\overline{RC}_j\right) + \sum_i \beta_i X^i_{j,t} + \epsilon_{j,t} \tag{3}$$

where $j \in \{1,...,7\}$ is the catchment index, $t$ is the time index, $i$ is the covariate index, and $\overline{RC}_j$ represents time-invariant, watershed-specific fixed effects (drainage area, land cover, mean climate), $X^i_{j,t}$ are the covariates, $\beta_i$ are regression coefficients

and $\epsilon_{j,t}$ is random noise. For covariates, we consider catchment-averaged October-May precipitation (P), relative humidity (RH), temperature (T), snow water equivalent (SWE) and snow fraction of precipitation (SF). Runoff has indeed been shown to be sensitive to these variables (e.g., Berghuijs et al., 2017). Indeed, enhanced precipitation or relative humidity lead to wetter soils and can be expected to be associated with higher RC values. Similarly, higher temperatures increase evapotranspiration and tend to decrease runoff. Finally, increased snow cover favors losses by sublimation and shifts the distribution of runoff

regimes towards slower runoff as opposed to rapid overland flow following rain storms. However, larger snow cover may also increase the risk of rain-over-ice events, which tend to have very high runoff coefficients (Davenport et al., 2020). Environmental model covariates are calculated using the ERA/MRCM run and associated snow model output. Runoff coefficients (RCs) for each catchment are defined as observed total October-June discharge divided by total October-June precipitation over the catchment. Precipitation is obtained from the ERA/MRCM run bias-corrected with TRMM. To assess the robustness of the

results to the choice of TRMM to bias-correct precipitation, we also calculate RCs based on precipitation series bias-corrected with the CHIRPS dataset (available from 1981-present).

Model selection is performed by stepwise regression: starting from a model with no covariates, covariates are added one at a time in the order of highest improvement to model skill, as determined by its Akaike Information Criterion (AIC). At each step, we also test whether removing any of the currently selected variables and replacing it by one of the remaining, non-selected

ones, brings any improvement. To estimate the sensitivity of RC to changes in climate conditions, we modify covariate values in the 1982-2011 ERA-Interim downscaled simulation by adding projected long-term changes in the GCM-driven simulations:

$$\log\left(\widehat{RC}_{j,t}^{m}\right) = \log\left(\overline{RC}_{j}\right) + \sum_{i\in\mathcal{I}}\beta_i\left(X_{j,t}^i + \overline{X}_m^i\right) \tag{4}$$

where $m \in \{1,2,3\}$ is model index, $\mathcal{I}$ is the set of optimal covariates, $X_{j,t}^i$ are ERA-Interim downscaled covariate values and $\overline{X}_m^i$ represent long-term covariate changes drawn at random according to:

$$225 \quad \overline{X}_m^i \sim N\left(\mu_{m,\mathrm{rcp}}^i - \mu_{m,\mathrm{hist}}^i, \sqrt{\sigma_{m,\mathrm{rcp}}^i + \sigma_{m,\mathrm{hist}}^i}\right) \tag{5}$$

where $\mu_{m,s}^i$ (respectively $\sigma_{m,s}^i$ is the average (respectively standard deviation) of covariate $i$ in model $m$ and scenario $s$. Results for the three models are then pooled together to yield a future distribution for $RC_{j,t}$.

## 4 Results

### 4.1 Snowpack modeling and projections

Annual cycles of reconstructed snow cover as a function of elevation are shown on Fig. 5. Overall, all models succeed in accurately reproducing snow cover dynamics in the region, although the ERA-Interim simulation tends to have a positive bias at high elevations, particularly above 3000 m (Fig. 5-b) and all simulations have a negative bias at low elevations (Figs. 5-d,e and 6-a,b). The GCM-driven experiments generally show too little snow cover, and a later snowpack build-up (Fig. 5-f). December-March average snow cover over the whole modeled area reaches 1460 km$^2$ in MODIS observations but only

1275 km$^2$ in the ERA-Interim driven run and 1185 km$^2$ in the ensemble mean historical average – a bias largely concentrated at low elevations (Fig. 6-a,b). Still, except for elevations below 2000 m, mean snow cover mostly remains close ($\pm$20%) to MODIS values. The inter-model spread in GCM-driven simulations also generally overlaps with observed snow cover values.

Consistent with these results, the spatial distribution of snow-to-precipitation ratio is generally well-reproduced in the GCM-driven experiments (Figs. 7-a,c and A1), apart from negatives biases at low elevations associated with the underestimation of snow cover (Fig. 6). Inter-annual variability in basin-wide snow cover is lower in the simulations compared to MODIS (standard deviation of 220-440 km$^2$ compared to 480 km$^2$ in MODIS) but the discrepancy is mainly due to the negative bias at low elevations, where snow plays a much more limited role in the overall water balance (T20a).

Still in the current climate, we find that annual relative sublimation losses are strongly linked to annual-mean precipitation, a relationship that the GCM experiments are able to capture (Fig. 8-a). Losses from latent heat fluxes are much smaller during wet years as compared to dry years, a relationship robust across all experiments. Wet years indeed bring higher RH over the region, due to enhanced moisture advection from the Atlantic which more than compensates for the larger heat advection and increased air temperatures that also occur in parallel (Knippertz et al., 2003). All experiments also agree reasonably well on the evolution of sublimation losses with elevation (Fig. 8-b). The MPI-ESM-MR and IPSL-CM5A-LR simulations exhibit a slight negative bias, but which remains within the uncertainty range in the assimilated run (T20a).

Future projections show a stark decline in snow cover across all the region (Figs. 6 and 9). The greatest relative decline is at low elevations, as expected since they are already seldom above the 0°C line (Boudhar et al., 2016). Above 2500 m, projections still exhibit a 30-40% decrease in snow cover area under RCP4.5 and 50-60% decrease under RCP8.5. Peak snow cover is reached around the beginning of February in all scenarios. However, snowpack build-up and melt occur respectively later and earlier in the season. The projected decline is even steeper in terms of snowpack water content (Fig. 10): SWE is reduced on average by 60% under RCP4.5 and 80-85% under RCP8.5, bringing peak SWE value from about 125 million m$^3$ (MCM) down to 20 MCM (Figs. 10-f, 11). Inter-model spread is largest in the RCP4.5 simulations, in which peak SWE ranges from 25-60% of its historical value.

## 4.2 Runoff modeling and projections

The panel regression and model selection framework are applied to runoff coefficients and selected covariates over the 1982-2011 period. Stepwise regression yields as optimal covariates relative humidity and snow fraction. The adjusted $r^2$ is equal to 0.30, meaning that these two covariates explain a small third of inter-annual variability in RC. Fitted RCs are shown against observed ones on Fig. 12. Consistent with the model $r^2$, fitted values have a much smaller variance. Still, for all the catchments, except Segmine, we observe a significant positive relationship between fitted and observed values. The coefficients for RH and snow fraction are both significant, with $\beta_{RH} > 0$ and $\beta_{SF} < 0$ (Table 3). In other words, all else being equal, a larger RH yields more runoff and more of the precipitation falling as snow yields less runoff. These results are robust to the choice of the precipitation dataset. When using the CHIRPS dataset, average RCs are generally lower, due to CHIRPS's wet bias compared to TRMM (Fig. 3); the optimal RC model includes RH and SF as well, but also precipitation as a third variable (Table 3). The values of $\beta_{RH}$ and $\beta_{SF}$ obtained with CHIRPS data are similar, although $\beta_{SF}$ is slightly less significant. In addition, for both precipitation datasets, the effect of RH on runoff coefficient, as measured by the respective regression coefficient $\beta_{RH}$, is 3-5 times that of snow. Since the input variables were not standardized prior to model fitting, this does not mean however that runoff coefficients are more sensitive to RH variations in the absolute.

Future runoff projections are characterized by consistent, steep declines in runoff coefficients of 5-17% under RCP4.5 and 15-30% under RCP8.5 (Fig. 13-a). The impact of decreasing RH largely dominates over that of declining snow fraction. Tarhat and Chacha, the two watersheds which already receive almost no snow in the present climate, exhibit the greatest relative RC decline, whereas in other watersheds, decreases in snow fraction help limit the decline in RC (Table 4). Combining now precipitation and runoff coefficient estimates, we find a 20-40% decrease in runoff in the RCP4.5 experiments, compared to a 50-65% decline under RCP8.5 (Fig. 13-b,c and Table 4). Decreases in precipitation drive most of the runoff trends, especially in the RCP8.5 scenario. Projected RC declines are about the same when using CHIRPS data for the analysis, although tend to be slightly higher, on average by 3-6% on average (not shown).

## 5 Discussion

### 5.1 Snowpack projections

The steep projected decline in snowpack in the High Atlas results both from a decrease in wet-season precipitation, and from warming trends which severely reduce the fraction of solid precipitation. This is particularly true at mid-elevations (2000-2500 m), very close to the zero-degree line in the current climate, where the relative decline in snow fraction is the highest (Fig. 7). At first order, changes in precipitation and rising temperatures each seem to account for about half of the projected SWE trends for the area as a whole. Precipitation goes down by about 25% under RCP4.5 and 40-45% under RCP8.5 (Tuel et al., 2020b), while the fraction of solid precipitation decreases by 25% under RCP4.5 and 50% under RCP8.5 (from $\approx$19% in the historical simulations to $\approx$14% and $\approx$9% annually in the RCP experiments) (Fig. 7). Rising temperatures prevent snowpack build-up not only by increasing the likelihood of liquid precipitation, but also by favouring snowmelt. Snowmelt is already frequent during winter in the High Atlas, even at high elevations (Tuel et al., 2020a), and is thus likely to be even more common in a warmer world. Still, at high elevations, the relative decline in snow fraction is roughly half as small, and consequently the precipitation signal may be dominant in the SWE trends.

Inter-model uncertainty in normalized SWE projections differs across scenarios (Fig. 11). It is small in the RCP8.5 simulations: peak SWE declines by 85$\pm$3%. In this scenario, warming is large enough in all the models that regardless of its magnitude, snowpack is projected to disappear almost entirely. All three models agree on a $\approx$ 50% decrease in snow-to-precipitation fraction. This is consistent with observations that in mountain regions, above-freezing temperatures are common, even at high altitudes, where average winter temperatures are not far from freezing (López-Moreno et al., 2017). Therefore, under the RCP8.5 scenario, the projected 4°C warming (Tuel et al., 2020b) would regularly bring all areas but the very highest peaks well above freezing, and prevent seasonal snowpack accumulation. By contrast, uncertainty is high in the RCP4.5 simulations, in which warming and drying are about half as small as under RCP8.5 (Table 4). Consequently, the reduction in the snow-to-precipitation fraction projected by the models ranges from 5 to 35%. In this scenario, the uncertainty in warming thus appears to be the main factor behind the spread in snowpack projections.

There is also uncertainty in absolute SWE values. As discussed in T20a, the historical SWE peak of 125 MCM is subject to caution; experiments with enhanced precipitation or reduced temperatures suggest it may in fact be as high as 200 MCM.

Still, while inter-model spread is large in the historical experiments (≈60 MCM) and still large (≈40 MCM) in the RCP4.5 experiment, it is reduced to almost zero under RCP8.5. Again, despite the spread in temperature and precipitation projections among the three models, they all agree on the virtual disappearance of the snowpack by the end of the century under business-as-usual.

## 5.2 Runoff modeling and projections

Among the several environmental variables tested, runoff coefficients in the High Atlas appear primarily affected by RH and the fraction of snow in seasonal precipitation. Consistent with previous studies (Wang et al., 2016; Duan et al., 2017), the higher RH is, the larger the runoff coefficient. Higher RH in the High Atlas is indeed associated with more precipitation, higher soil moisture levels and a lower evaporative demand. Thus higher RH will lead to reduced relative sublimation losses (Fig. 8-a), but it is also linked to warmer temperatures which tend to decrease the snow-to-precipitation ratio and lead to lower
relative sublimation losses as well. RH is therefore important in our model because it incorporates information on both the energy constraint (evaporative demand) and on the water availability. The snow fraction has an opposite effect: the larger it is, the lower runoff efficiency. We can understand the influence of snow fraction by noting that a higher snow fraction means more opportunity for sublimation, particularly large at high elevations, and evaporation of melted snow, consistent with our analysis of sublimation losses (Fig. 11). Precipitation in the area tend to occur in short and intense storms, and quickly saturate
the dry soil, leading to rapid overland flow with limited opportunity for evaporation (El Khalki et al., 2018). By comparison, snowmelt is slow and leads to a more gradual surface flow with the potential for higher evaporative losses in this climate where evaporation tends to be water-limited. More winter snow may also lead to a higher likelihood of rain-on-snow episodes in spring, known to cause rapid flooding due to high runoff efficiencies (Davenport et al., 2020). Whether this occurs in the High Atlas remains to be shown, but it would nevertheless be consistent with the effect of the snow fraction on runoff efficiencies.
More snow may also mean more opportunity for infiltration and aquifer recharge (Hssaisoune et al., 2020). It should be noted that karstic areas are also quite frequent within the Oum-Er-Rbia watershed (Akdim, 2015), with important implications for infiltration, aquifer and spring regimes in our study area. Compared to surface runoff, groundwater remains a small fraction of water use in the Oum-Er-Rbia basin (<15%), and even less of available renewable water since aquifers are largely overdrawn (Hssaisoune et al., 2020). Groundwater data are quite scarce in this region; we make the choice of purely focusing on surface
runoff, keeping in mind that a more complete picture of basin-wide water availability would also require taking aquifer fluxes into account. While base flow is negligible in six out of the seven sub-catchments considered here, aquifer discharge may still occur naturally further down the mountains, or artificially via direct groundwater pumping in the agricultural plains.

Results indicate a decline in runoff under future climate scenarios, due to a decrease in both runoff efficiency and wet-season precipitation. Precipitation trends play a much larger role than RC changes in runoff projections. In RCP8.5, precipitation
decreases by 48-57%, compared to a 15-30% decline in RCs. Our projected runoff trends are consistent with those of El Moçayd et al. (2020) who focused on catchments part of the Sebou watershed, where snow plays a much smaller role than in our region (Marchane et al., 2015). This is not surprising given the weak impact of the declining snow fraction on runoff coefficients, compared to that of the relative humidity decline. Runoff efficiency is projected to decline mainly due to atmospheric drying

(decreasing RH), which more than compensates for the diminishing snow fraction. Under RCP8.5, average relative humidity will decline by 3-6% (Tuel et al., 2020b), which will lead to an average 30% RC decline across the seven sub-catchments, while the decline in snow fraction increases RCs by only 10%. Additionally, lower RH implies that sublimation rates will be higher when snow is present. However, because the snow-to-precipitation ratio will also sharply decline due to rising temperatures, the overall loss of annual precipitation by sublimation will tend to decrease by about a third (Fig. 8-b).

The spread in runoff projections is large (Fig. 13-b), due to uncertainties in both RC (Fig. 13-a) and precipitation projections. Despite warming and drying trends being smaller in RCP4.5, the uncertainty on RC changes is just as high as under RCP8.5 (Fig. 13-a), notably because snowpack projections are much more uncertain (Fig. 11). Combined with weaker precipitation changes, this translates into larger uncertainty in future runoff under RCP4.5 compared to RCP8.5.

## 5.3 Uncertainties in results and limitations of the study

Every modelling framework relies on its own assumptions, which translate into uncertainties. First, it is important to note that our pixel-by-pixel approach to bias-correction does not take spatial correlation and inter-variable dependencies into account, for which a more complex method like R2D2 (Vrac, 2018) would be required. However, in the present case, it makes little practical difference. To fit the snow model, we rely on average annual cycles of snow cover over which short-term correlations across space and between variables, beyond the seasonal cycle, have little influence. In addition, the target datasets used in the bias-correction come from different sources (TRMM, MODIS, etc.), and thus we should be very careful about relying on their correlations when correcting the data. Ideally, one could correct all variables from a single observational dataset that includes all the variables we use, but such a dataset obviously does not exist.

Second, for future climate projections, we rely on a single regional climate model, and thus do not explore the full range of uncertainties related to model configuration and parametrizations. Still, the regional simulations we use here have been specifically tailored to the area, particularly the choice of driving GCMs, and validated against a range of observations (Tuel et al., 2020b). The fact that the uncertainty in snowpack projections under RCP8.5 is small could be further explored by using other GCM/RCM combinations, especially ones that lead to less warming than projected in our three-member ensemble. As to the uncertainty in precipitation trends, it is difficult to reconcile. As shown by Tuel and Eltahir (2020), the magnitude of future wet-season precipitation in Northwestern Africa is mainly determined by that of changes in Mediterranean atmospheric circulation. Spread in dynamical trends is difficult to reduce for this region. A GCM-selection approach based on storylines could be relevant to determine plausible scenarios (Shepherd, 2019). In addition, drier soils together with enhanced absorption of solar energy where snowpack disappears will also lead to enhanced warming locally, driving yet further snowpack melt. We do not explicitly take this into account in our model. In particular, the snow albedo effect is largely absent from the MRCM simulations due to their 12 km resolution, which is still too coarse to represent the complex topography. For areas at the highest elevations (near 4000 m) which will still likely be below freezing in future winters, melt may remain largely unchanged in the middle of winter; however, a drier atmosphere will still be associated with reduced precipitation and increased sublimation losses, which will play a critical role in reducing the snowpack.

The advantage of using a statistical regression model to investigate the sensitivity of runoff to climate variables and its changes

is its simplicity and interpretability. It allows for instance to highlight the influence of the snow fraction in the inter-annual variability of RCs. While hydrological models are certainly more comprehensive, they also rely on many parameters which leaves them vulnerable to overfitting and thus to even more uncertainties. In addition, they suffer from the same drawback as our simple model, which is that there is no guarantee that the relationships obtained in the current climate are directly transferable to the future climate. Still, the large inter-annual variability in climate variables in the present climate helps make the model more robust to large changes in average covariate values. Furthermore, the explanatory variables used in the regression are physical variables which we understand how mechanistically relate to runoff. The difference with a hydrological model is in the different form of mathematical relations, however the physical relationship between the variables is somewhat respected. Furthermore, RH and snow fraction only explain 30% of inter-annual variability in RCs. The rest of the variability may be still explained by climatic variables, which we do not include in our model. Our seasonal approach to defining predictor variables is also quite restrictive; it does not take into account the weekly variability (in RH, temperature, precipitation, etc.) which may affect runoff efficiency. For instance, the details of the daily distribution of precipitation are important to estimate runoff: the more concentrated in time precipitation is, the less opportunity for infiltration and evaporation, and therefore the higher the runoff efficiency.

Finally, we focus on climate-driven inter-annual variability in RCs; thus, we do not explain differences in average RCs across catchments, nor can we say anything about long-term variability in RCs forced by non-climatic parameters, like land use change. This last aspect may not be critical in the case of the High Atlas since most of the area under study is uncultivated, naturally lacking tree cover and sparsely populated. It has thus not experienced very significant land use changes in the last few decades. However, climate-change driven trends in vegetation cover may still affect runoff efficiency in the region. It is also unclear to what extent enhanced groundwater pumping in the Oum-Er-Rbia watershed since the 1980s may have modified the natural water balance and, indirectly, RCs in mountain catchments.

## 6   Conclusions

Based on the robust understanding of its snow water balance in the current climate, we quantified in this study the response of the High Atlas snowpack to climate change using high-resolution downscaled climate projections. We find that the High Atlas snowpack will significantly decline, even in the RCP4.5 scenario which involves substantial mitigation of emissions. By the end of the century, snow may become a rarity below 2000 m. Peak snow cover is projected to go from 17% of our study area down to 9% under RCP4.5 and even 4% under RCP8.5. In parallel, snowpack water equivalent could decline by 80%, even at the highest elevations (>3500 m). Snowpack decline is evidently connected to regional warming trends, but also affected by a projected 40-60% decrease in wet-season precipitation in Northwestern Africa.

The analysis of runoff coefficients for seven mountain catchments showed that a third of their inter-annual variability could be explained by large-scale meteorological factors like snow fraction of precipitation and relative humidity. Interestingly, in this region, a larger snow fraction leads to lower runoff coefficients. While the reverse is generally thought to be true at higher latitudes (Berghuijs et al., 2014), this finding is consistent with other analyses in warm, semi-arid regions that receive substantial

amounts of snow during winter (Davenport et al., 2020). Warmer conditions tend to enhance runoff efficiencies by reducing snowpack, thus limiting sublimation losses and the slow melting of snow, propitious to evaporation, and by increasing the likelihood of rain-on-snow events that tend to cause high runoff efficiencies. While decreasing snowfall will partly compensate for the projected atmospheric drying over the region, runoff coefficients will tend to decline by 5-30% depending on catchment and scenario. Combined with precipitation trends, basin-wide runoff could be reduced by 60% in the worst case.

The robust physical understanding behind large-scale projections for the region (Tuel and Eltahir, 2020) increases the likelihood that the dire projections detailed above will be realized provided greenhouse gas emissions are not brought under control. This would deal a severe blow to the region, jeopardizing its agriculture-based economy and the livelihood of millions of smallholder farmers. Agriculture, which accounts for 90% of current water use, will have no choice but to adapt. A 40-60% precipitation decline would make rainfed agriculture infeasible. At the same time, availability of water for irrigation will also decline sharply. A change of cropping patterns, like a transition to tree crops with less water demand and higher economic value like olives, will likely be unavoidable for Morocco to adapt to this future reality.

*Data availability.* ERA-Interim reanalysis data are available from https://apps.ecmwf.int/datasets/. TRMM data are available from https://disc.gsfc.nasa.gov/datasets/TRMM_3B42_Daily_7/summary, and CHIRPS data from https://data.chc.ucsb.edu/products/CHIRPS-2.0/. CMIP5 model output were downloaded from https://esgf-index1.ceda.ac.uk/projects/cmip5-ceda/. MRCM simulations used in this study are available from the corresponding author upon request.

# Appendix A: Appendix

## A1 Supplementary figures A1 and A2

*Author contributions.* EABE conceived and supervised the study. AT carried out analyses and wrote the first manuscript draft. NEM and MDH contributed with data and interpretation of results. All authors contributed to editing the manuscript.

*Competing interests.* The authors declare that they have no competing financial interests.

*Acknowledgements.* The authors would like to thank Michael Follum and Jeffrey Niemann for providing their RTI-SNOW-17 code.

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

**Table 1.** Overview of datasets used in this study.

| Data | Description | Availability |
|---|---|---|
| Station precipitation | Daily precipitation measured at seven locations in the Oum-Er-Rbia watershed | 1980-2015 |
| Station discharge | Daily discharge measured at seven locations in the Oum-Er-Rbia watershed | 1978-2015 |
| TRMM TMPA 3B42 version 7 | Satellite-based 3-hourly precipitation at 0.25° resolution | 1998-present |
| CHIRPS v2.0 | Satellite- and station-based 6-hourly precipitation at 0.05° resolution | 1981-present |
| MODIS Land Surface Temperature L3 version 6 (MOD11A1) | Satellite-based land surface temperature at 1 km resolution | 2000-present |
| MODIS Terra snow cover daily L3 (MOD10A1) | Satellite-based fractional snow cover at 500 m resolution | 2000-present |
| ERA/MRCM | Regional downscaling of ERA-Interim with MRCM at 12 km resolution (from Tuel et al. (2020b)) | 1981-2011 |
| ERA/GCM | Regional downscaling of three CMIP5 GCMs (IPSL-CM5A-LR, GFDL-ESM2M and MPI-ESM-MR) with MRCM at 12 km resolution (from Tuel et al. (2020b)) | 1976-2005 (historical) and 2071-2100 (RCP4.5 and RCP8.5) |
| SRTM 90-meter resolution version 4.1 (STRM90) | Satellite-based elevation at 90 m resolution | N/A |

**Table 2.** Characteristics of the seven analyzed sub-basins.

| Name | Area (km$^2$) | Mean elevation (m) |
|---|---|---|
| Tarhat | 997 | 1627 |
| Chacha | 1519 | 1460 |
| Ouchene | 2391 | 1953 |
| Tillouguite | 2488 | 2363 |
| Moulay Hassan | 1700 | 2124 |
| Segmine | 506 | 1897 |
| Tamesmate | 1303 | 2198 |

**Table 3.** Runoff coefficient model results. The bottom three lines show coefficient values (left-hand column) and their statistical significance (p-value, right-hand column).

|  | MRCM-BC | | CHIRPS | |
|---|---|---|---|---|
| $r^2$ | 0.30 | | 0.36 | |
| SF | -1.84 | 4E-03 | -1.47 | 2E-02 |
| RH | 5.71 | 2E-05 | 8.71 | 1E-09 |
| Pr | – | – | 2.04E-03 | 1E-06 |

**Table 4.** Long-term (2071-2100 minus 1976-2005) projections for the seven catchments: October-May temperature (°C), precipitation (%), relative humidity (%), absolute and relative snow fraction (%), runoff coefficient (%) and runoff (%), under RCP4.5 and RCP8.5.

| Catchment | M. Hassan | | Tamesmate | | Tillouguite | | Segmine | | Ouchene | | Tarhat | | Chacha | |
|---|---|---|---|---|---|---|---|---|---|---|---|---|---|---|
| | RCP4.5 | RCP8.5 | RCP4.5 | RCP8.5 | RCP4.5 | RCP8.5 | RCP4.5 | RCP8.5 | RCP4.5 | RCP8.5 | RCP4.5 | RCP8.5 | RCP4.5 | RCP8.5 |
| $\Delta T(°C)$ | 2.3 | 4.6 | 2.2 | 4.4 | 2.2 | 4.5 | 2.3 | 4.6 | 2.3 | 4.6 | 2.3 | 4.8 | 2.3 | 4.7 |
| $\Delta$Pr (%) | -11 | -53 | -14 | -57 | -30 | -57 | -22 | -48 | -31 | -57 | -31 | -50 | -31 | -55 |
| $\Delta$RH (%) | -3.1 | -6.2 | -2.5 | -5.5 | -3.2 | -6.4 | -3.6 | -7.1 | -3.4 | -6.3 | -3.2 | -6.6 | -3.2 | -6.1 |
| $\Delta$Snow (%) | -3.3 | -11.3 | -3.9 | -12.3 | -6.7 | -15.7 | -5.7 | -14.9 | -6 | -11.3 | -5 | -7.5 | -2 | -3.2 |
| $\Delta$SF (%) | -14 | -46 | -16 | -51 | -19 | -44 | -19 | -51 | -28 | -52 | -52 | -78 | -50 | -80 |
| $\Delta$RC (%) | -14 | -23 | -8 | -17 | -6 | -14 | -10 | -17 | -9 | -19 | -13 | -27 | -15 | -30 |
| $\Delta$Runoff (%) | -23 | -64 | -21 | -64 | -34 | -63 | -29 | -57 | -37 | -65 | -39 | -63 | -41 | -68 |

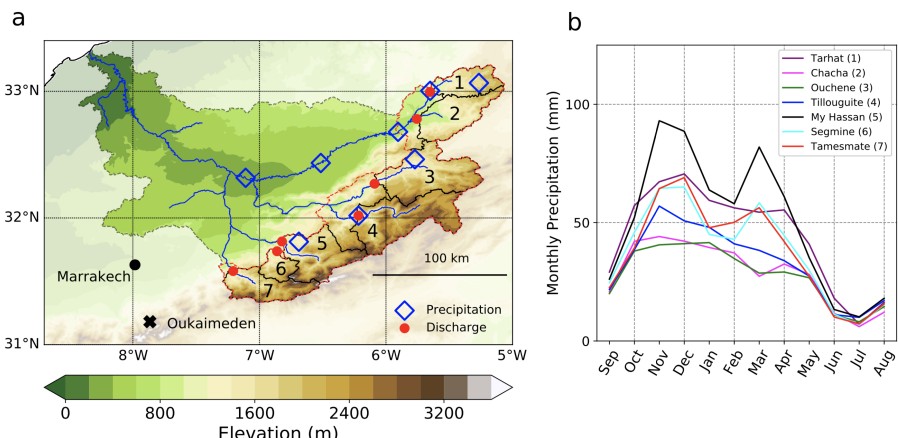

**Figure 1.** (a) Map of the Oum-Er-Rbia watershed, with elevation shown by shaded contours. The main waterways are indicated by solid blue lines. Blue diamonds and red circles indicate the location of precipitation and river discharge stations, respectively. The snow modeling domain is shown by the dashed red line. The seven catchments defined by the discharge stations are shown by solid black lines and are indicated by numbers: (1) Tarhat, (2) Chacha, (3) Ouchene, (4) Tillouguite, (5) Moulay Hassan, (6) Segmine and (7) Tamesmate. The location of the Oukaimeden snow station, outside our study area, is shown by a black cross. (b) Annual cycles of precipitation for the seven catchments, based on TRMM data (1998-2015).

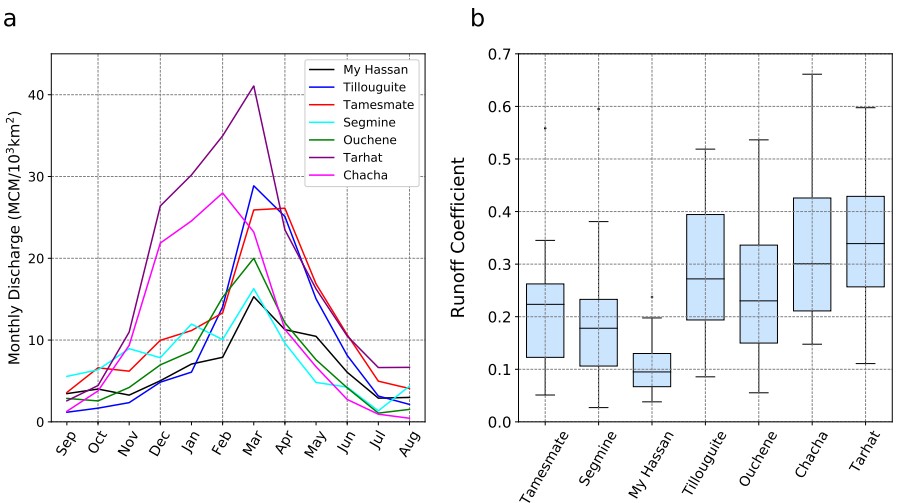

**Figure 2.** (a) Annual cycles of monthly runoff at the seven runoff gauges, after base flow removal and normalization by catchment area (km$^2$). (b) Boxplot of annual runoff coefficients for the seven catchments (1982-2011), using ERA/MRCM precipitation bias-corrected with TRMM data.

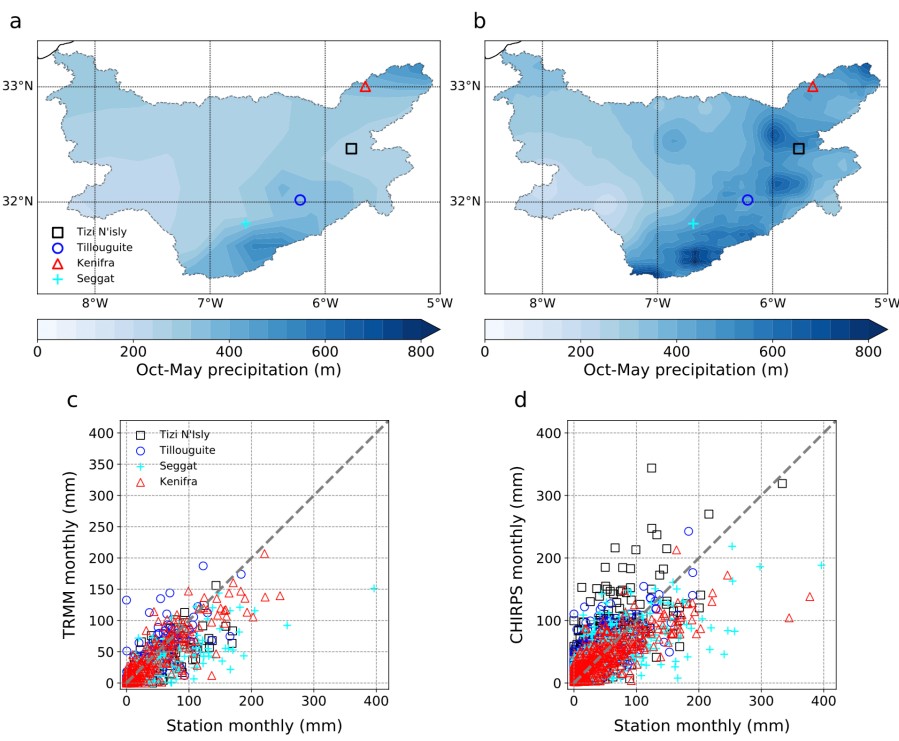

**Figure 3.** (a) October-May average precipitation (mm) over the Oum-Er-Rbia watershed, from TRMM (1998-2015). The four available precipitation stations above 1000 m elevation are indicated by symbols. (c) Monthly precipitation at the four stations shown on (a) against corresponding TRMM values. (b,d) Same as (a,c) but for the CHIRPS dataset (1981-2015).

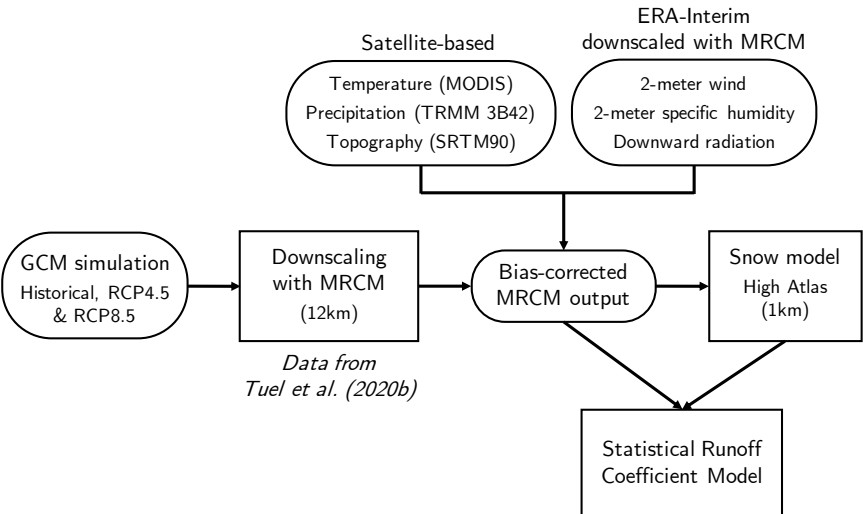

**Figure 4.** Summary of methodology and input datasets used to assess climate change impacts on snowfall, snowpack and runoff in the Oum-Er-Rbia watershed.

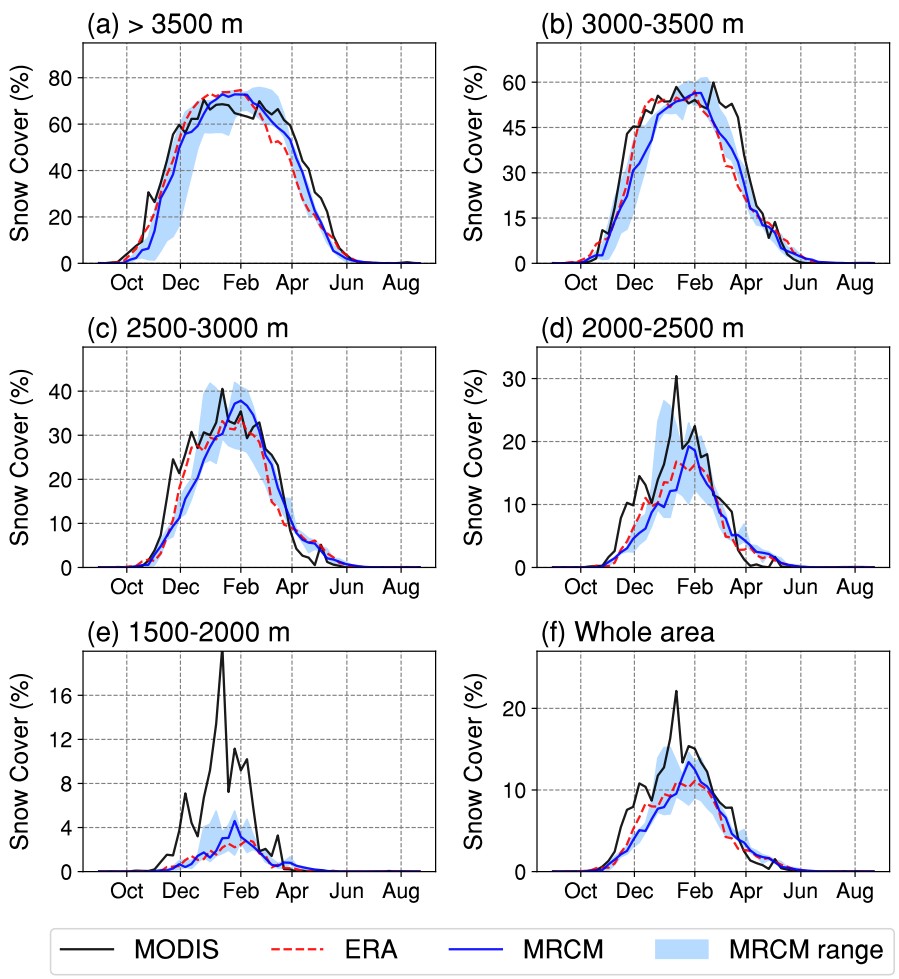

**Figure 5.** Annual cycles of snow cover (in %) in the MODIS observations (black), ERA- Interim simulation (dashed red) and three GCM-driven historical simulations (solid blue: median; blue shading: 3-model range), at various elevations ranges within our study area: (a) > 3500 m, (b) 3000-3500 m, (c) 2500-3000 m, (d) 2000-2500 m, (e) 1500-2000 m and (f) whole area.

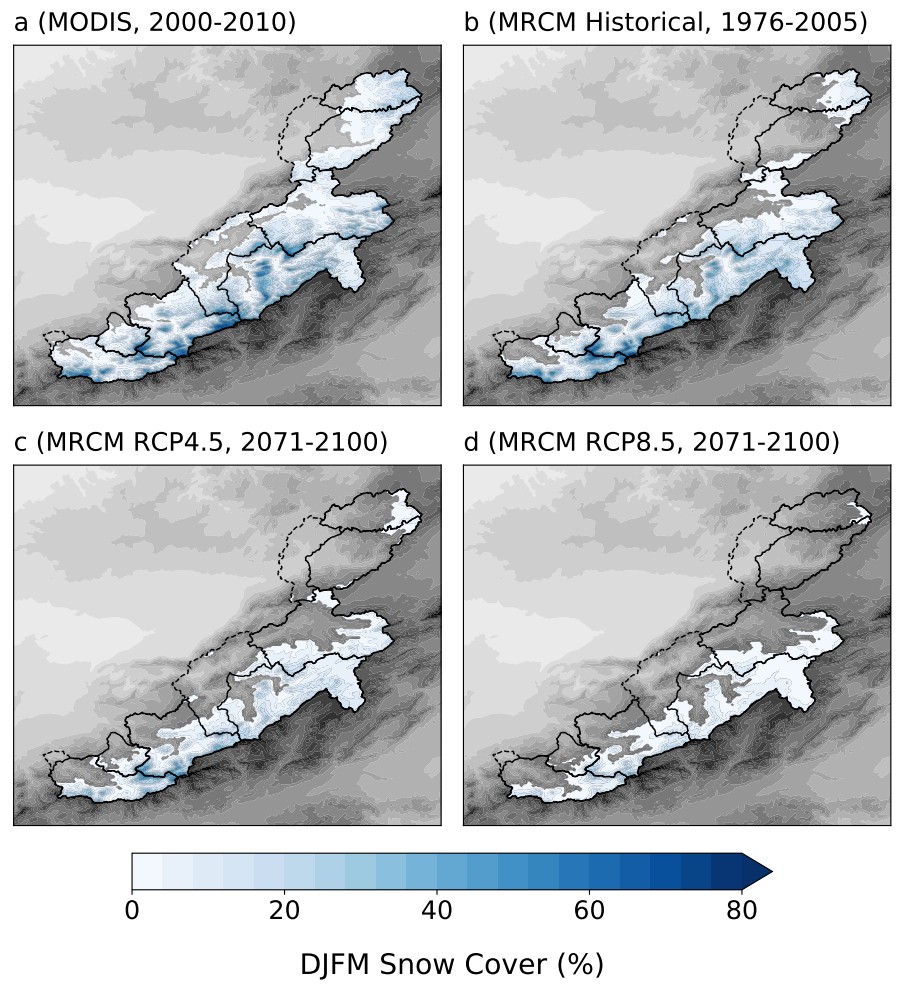

**Figure 6.** Mean December-to-March (DJFM) fractional snow cover (%) over the basin in (a) MODIS (2000-2010) data, and (b-d) three-GCM average under the (b) historical (1976-2005), (c) RCP4.5 (2071-2100) and (d) RCP8.5 (2071-2100) experiments.

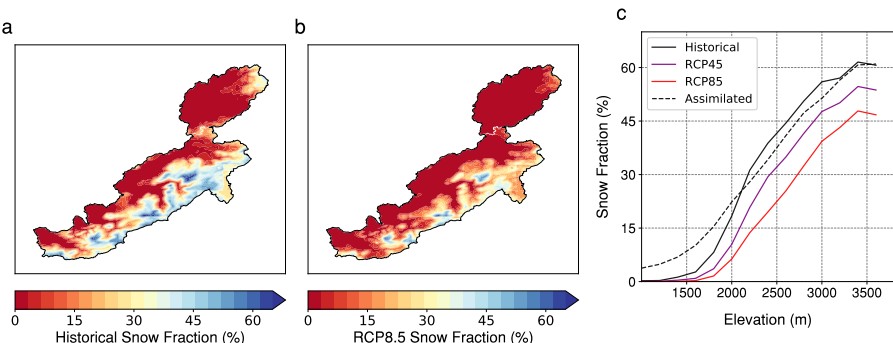

**Figure 7.** (a-b) Snow fraction of annual precipitation in the (a) historical and (b) RCP8.5 scenario (average between all three GCM-driven simulations). (c) Snow fraction of annual precipitation as a function of elevation, in each scenario (historical, RCP4.5 and RCP8.5; three-model average) and in the assimilated control simulation of T20a.

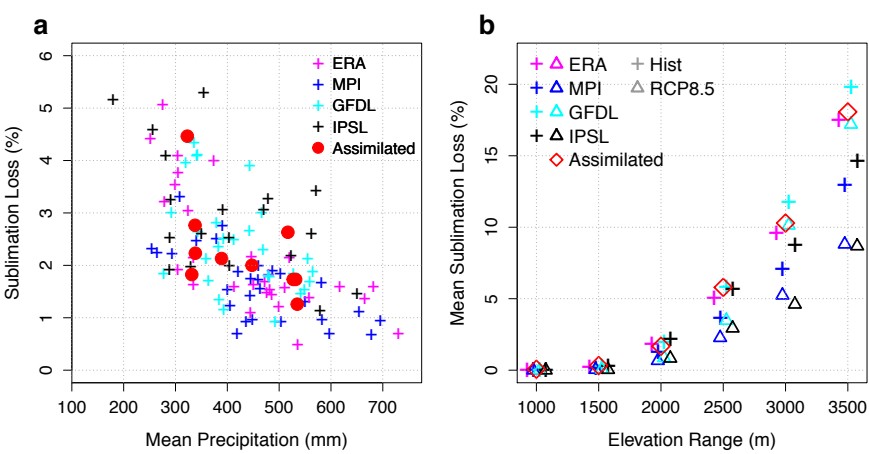

**Figure 8.** (a) Fraction of annual precipitation lost by sublimation against total annual precipitation in the MRCM downscaled experiments forced with ERA-Interim (1982-2011, magenta), the three GCMs (1976-2005): MPI-ESM-MR (blue), GFDL-ESM2M (cyan) and IPSL-CM5A-LR (black), and the assimilated snow run forced with MODIS and TRMM data only (red, data from T20a). (b) Fraction of annual precipitation lost by sublimation as a function of altitude range in our study area, for the assimilated run from T20a and in the various downscaled experiments, for historical (1976-2005, "+") and RCP8.5 (2071-2100, "Δ") scenarios.

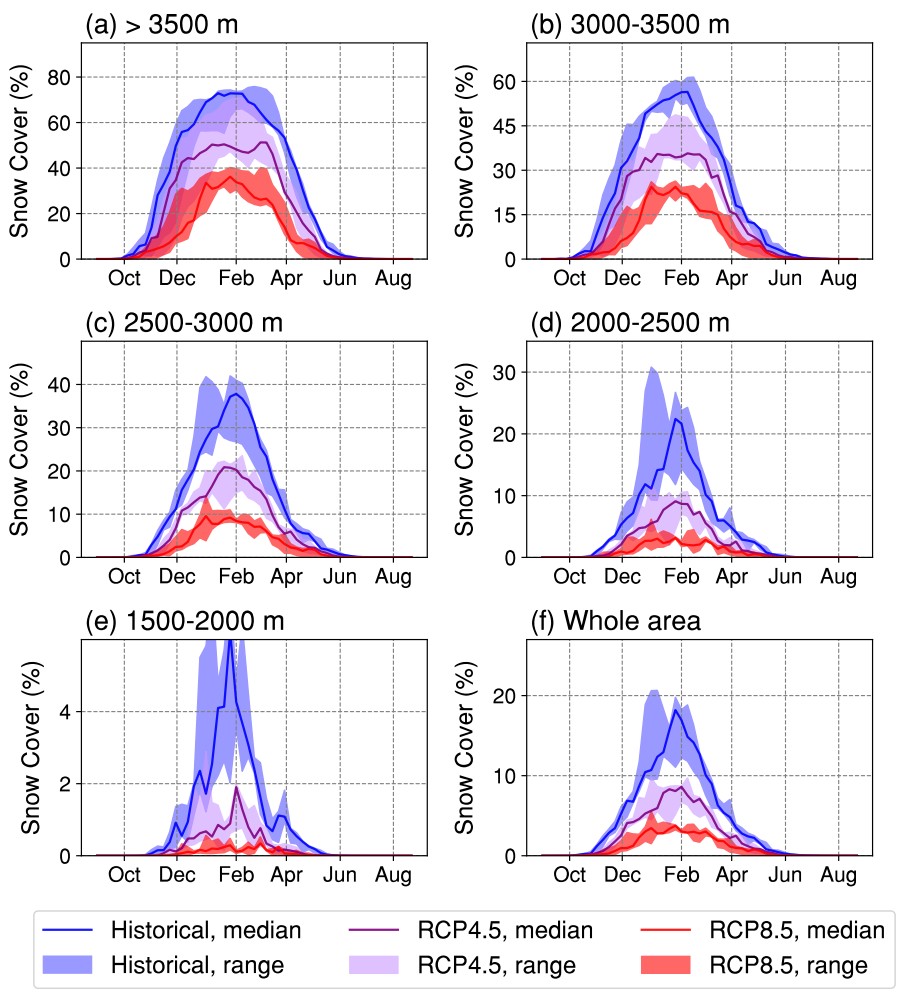

**Figure 9.** Annual cycles of snow cover (in %) in the three GCM-driven experiments under the historical (blue, 1976-2005), RCP4.5 (purple, 2071-2100) and RCP8.5 (red, 2071-2100) scenarios, at various elevations ranges within our study area: (a) > 3500 m, (b) 3000-3500 m, (c) 2500-3000 m, (d) 2000-2500 m, (e) 1500-2000 m and (f) whole area. Solid lines represent the three-model medians and the shading corresponds to the three-model spreads.

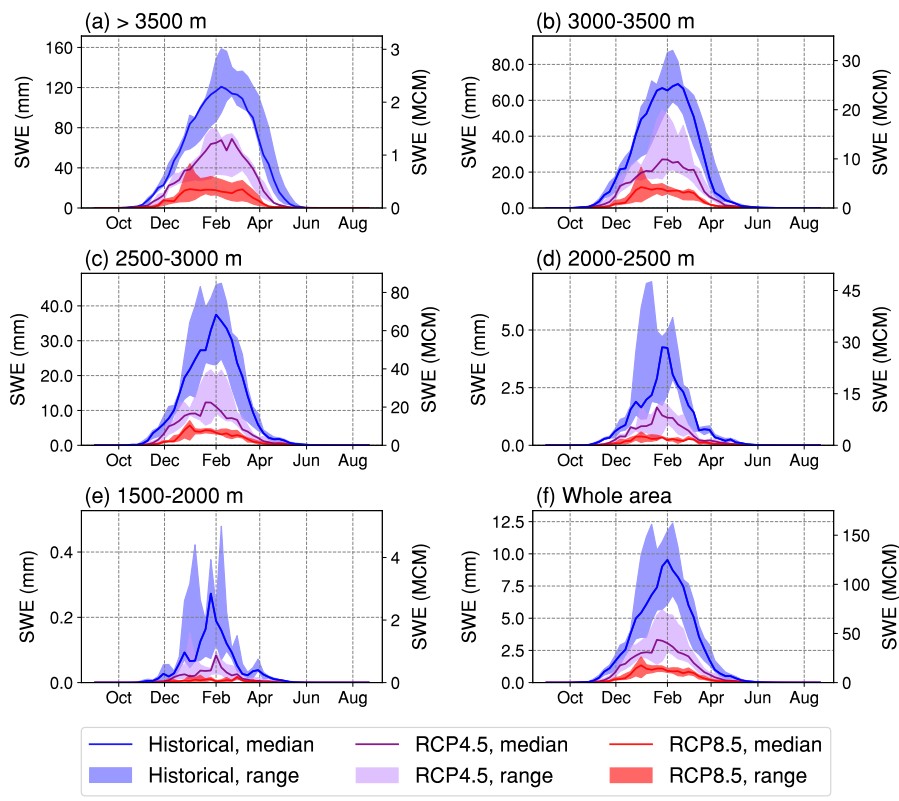

**Figure 10.** Same as Fig. 9, but for average snow water equivalent (mm, left-hand axis) and corresponding total snow water content (million m$^3$, MCM).

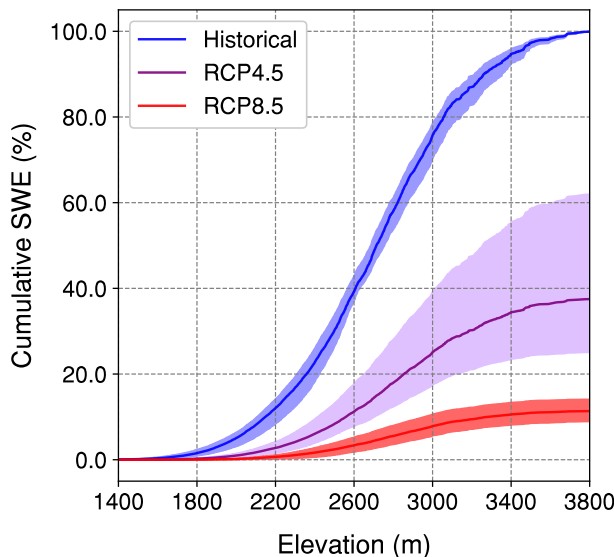

**Figure 11.** Distribution of cumulative basin-wide peak SWE with elevation in the GCM-driven experiments, under the historical (black), RCP4.5 (purple) and RCP8.5 (red) scenarios. SWE is normalized in each model by that model's historical total basin-wide SWE.

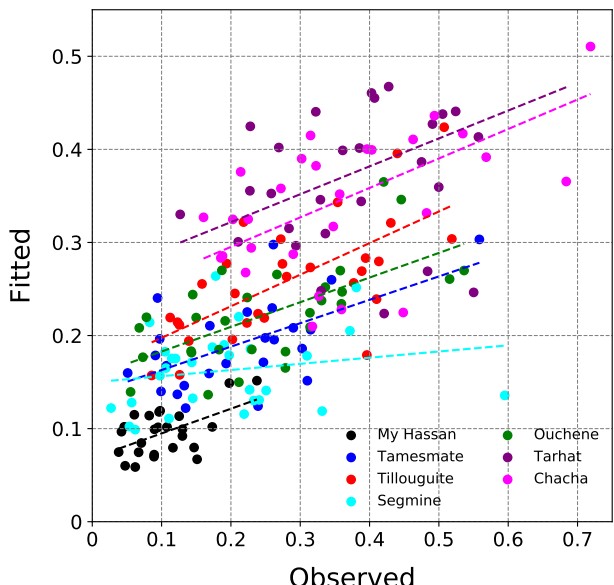

**Figure 12.** Fitted runoff coefficient values against observed values (defined with TRMM precipitation), for the seven catchments in our study area. Best-fit linear regression lines are shown by dashed lines.

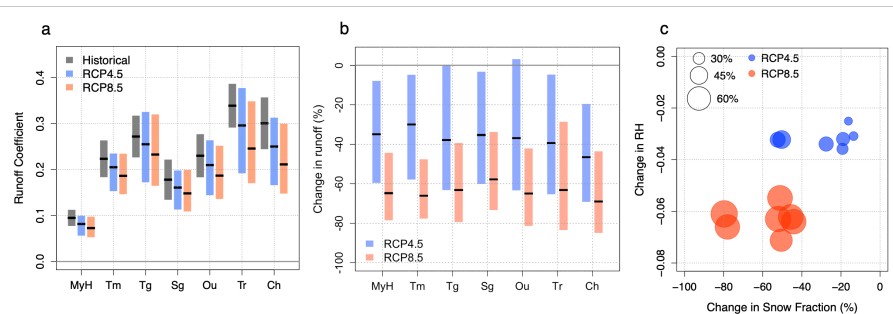

**Figure 13.** (a) Average runoff coefficients for the seven catchments in the observations (black), and projected average values in the RCP4.5 (blue) and RCP8.5 (red) scenarios. Boxes represent 90% confidence intervals. (b) Projected relative changes in runoff across the seven catchments against relative change in snow fraction (x-axis) and change in catchment-wide relative humidity (y-axis), for the RCP4.5 (blue) and RCP8.5 (red) scenarios.

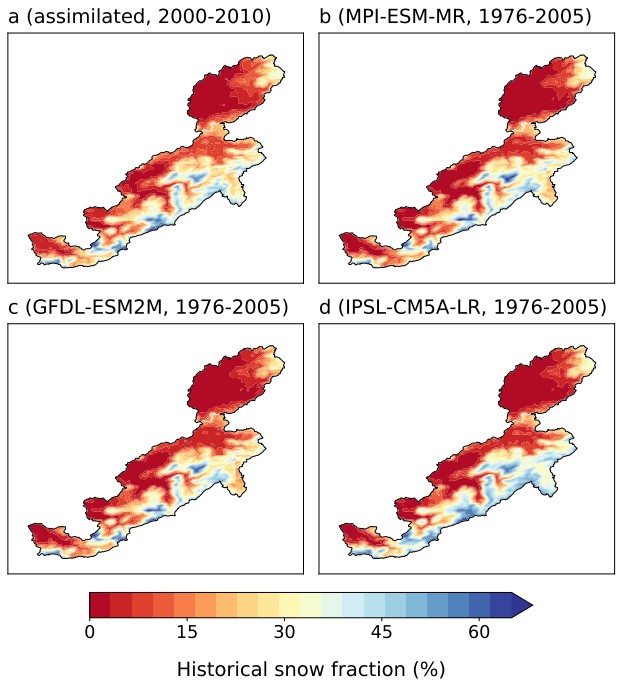

**Figure A1.** Snow fraction of annual precipitation in (a) the assimilated run of Tuel et al. (2020a) (2000-2010), and (b-d) the historical (1976-2005) runs of each of the three selected GCMs.

**Table A1.** List of precipitation stations.

| Name | Lon (°E) | Lat (°N) | Elevation (m) |
|---|---|---|---|
| Dechra El Oued | -5.90 | 32.68 | 595 |
| Kenifra | -5.65 | 33.00 | 1036 |
| Mechra Eddahk El Oued | -6.52 | 32.43 | 406 |
| Ouled Sidi Driss | -7.11 | 32.32 | 320 |
| Seggat | -6.69 | 31.81 | 1150 |
| Tillouguite | -6.22 | 32.02 | 1100 |
| Tizi N'Isly | -5.77 | 32.46 | 1595 |