# Peer review of "Future projections of High Atlas snowpack and runoff under climate change"

_Hydrology and Earth System Sciences, 2020_

## Referee Comment (RC1) · Anonymous Referee #1 · 4 Jan 2021

The manuscript of Tuel et al is well written and pleasant to read, it deals with the impacts of climate change on the snow cover in Morocco which is very important for surface water resources. I think this is an interesting contribution for HESS. The main problem with this manuscript is a partial description of the methods used, so it is difficult to say whether certain assumptions, for example on bias-correction, may have a large impact on the results. See my specific comments below.

One element that needs to be discussed in the manuscript is that the runoff coefficients are not only impacted by climatic parameters but also by surface conditions. Significant changes in vegetation cover, land use or agricultural practices often have a greater impact, and this aspect is absent from the manuscript.

Page 1, line 30 : no reference to the litterature except to the previous

publication of the authors, while an important body of work exist for the Mediterranean and Morocco : Driouech et al https://doi.org/10.1007/s41748-020-00169-3, Drobinski et al https://doi.org/10.1007/s10113-020-01659-w, Cramer et al https://doi.org/10.1038/s41558-018-0299-2, Lionello et al https://doi.org/10.1007/s10113-018-1290-1

In particular, climate projections on runoff already exist for the basin of interest, see Jaw et al https://doi.org/10.1016/j.ejrh.2015.02.008 Tramblay et al http://doi.org/10.1007/s11269-017-1870-8

Page 2, line 38 : what is a "parametric snow module" ?

Section 2.2: it is not clear why the authors consider TRMM rainfall, while daily precipitation data is available at 7 locations (line 75). Why also mention CHIRPS rainfall if it is not used in the study, as the author state

Section 2.3: The authors should justify why they rely on only one RCM, when nowadays large ensemble of climate model experiments are available, such as the Euro-CORDEX (Jacob et al https://doi.org/10.1007/s10113-020-01606-9) or Med-CORDEX (Ruti et al https://doi.org/10.1175/BAMS-D-14-00176.1) initiatives.

It is well established that to obtain robust projections it is necessary to consider several combinations of GCM/RCM, and the use of only one RCM strongly reduce the relevance of the work (see Frernandez et al https://doi.org/10.1007/s00382-018-4181-8).

In addition, the RegCM version 3 is rather outdated (2006) since the current version is RegCM-4 (https://www.ictp.it/research/esp/models/regcm4.aspx).

Page 5, line 142: the bias correction method is not detailed. what kind of approach is used beside the use of CDFt? In such a mountainous area, and since this study consider several variables in RCM simulations (temperature, precipitation, humidity...) a pixel-by-pixel and variable-by-variable bias correction with CDFt without considering the spatial correlation and inter-variable dependencies can lead to strong uncertainties.

See Vrac et al https://hess.copernicus.org/articles/22/3175/2018/

It is quite surprising that the authors seems to apply a simplistic method for bias correction of RCM outputs while they develop a high-resolution physically based framework for snow simulations.

Page 6, line 178: similar to my comment above, how do you compute catchment-averaged October-May precipitation ? with observed rainfall or TRMM ?

Page 6, line 177: watershed-specific fixed effects, are the parameters fixed according to size, land use etc. ?

Page 6, line 180: It is not clear if these sentences are results of sensitivity analysis or results of previous works

Page 6, line 185: Only precipitation and temperature are bias-corrected ? Line 125 the author state they use 6-hourly wind speed, specific humidity, air temperature, precipitation, and downward longwave and shortwave from the RCM simulations. Later on in the text, relative humidity seems to be an important driver of change, therefore better explanations on the method used to bias correct this parameter (and others) are required

Page 7, line 188, I don't understand this sentence "Therefore, we use the ERA/MRCM precipitation data, bias-corrected with TRMM,"

Page 8, line 213: the author mention a "statistical downscaling of the MRCM output to 1km,", but nothing in the method section about this. How is it possible to downscale 12km RCM simulations to 1km, with "reference" precipitation being TRMM data at 25km spatial resolution ? there is a confusion here between "downscaling" and "bias correction".

Section 4, results I see no validation of the methods applied, prior to produce future scenarios. What about the efficiency of bias-correction ? What about the efficiency of panel regression to reproduce the inter-annual variability of runoff coefficients ? since

the authors rely on TRMM rainfall, it would be interesting to see a comparison of the model driven by either observed rainfall or TRMM to reproduce discharge dynamics

Page 10, line 280: it should be noted that the Oum Rbia basins has several areas with karstic functioning

Page 10, line 294-297: this is not a result and should be in the introduction. The "Source: Direction de la Recherche et de la Planification de l'Eau, Rabat" is not in the reference list. This is not a result of the present study since the data and method used to obtain this result are not presented

---

## Referee Comment (RC2) · Anonymous Referee #2 · 8 Jan 2021

Future projections of High Atlas snowpack and runoff under climate change Tuel et al. 2020: https://doi.org/10.5194/hess-2020-622

General comment

Tuel et al. simulate the snow cover of part of the High Atlas mountains in Marocco (headwater catchments of the Oum-Er-Rbia watershed) using downscaled and bias corrected climate model data (data from three GCMs and two RCPs). Furthermore, they use a statistical runoff coefficient model to estimated changes in runoff. They concluded that decreases in precipitation in combination with increases in temperatures will diminish snowpacks and reduce runoff. In general, it is an interesting study and has the potential to become a valuable contribution to hydrological research. However, I see several major issues regarding the analytical approaches and the text that need

to be addressed before it can be considered for publication at HESS. It needs to become more clear what are the new findings of this study that extend already existing knowledge, I think.

Major comments

Comment 1: Language

Although I am not a native English speaker myself, I find that the language of the manuscript needs to be improved. I came across numerous sentences and expression that I think need improvement (see specific comments). In my opinion, a language check will improve the manuscript. Furthermore, make sure to use a space to separate the unit from the number.

Comment 1: Structure text

A restructuring of the text can increase its comprehensibility. I think, it would be helpful to separate Results and Discussion. First present results with subsections on snow simulations and runoff coefficient model and then discuss results in a next step (maybe again use same subsection, e.g. 'snow simulations' and 'runoff coefficient model'.

Comment 3: Introduction

I miss a clear line of argumentation here that then leads to your aims/goals. Please restructure and rephrase so that the literature and numbers presented are easier to understand (see also specific comments). I think the main sentence is in Page 2 Line 39-40. This is your main motivation, right?

Comment 3: Data

This section was a bit confusing for me. You present so many different data sets with different time frames, observed, simulated, station-based, satellite-based,..Often I did not understand what you use the individual data sets for. Please support the reader by e.g. an overview scheme and/or table that illustrates what data sets you use, what

time frame and what for. As far as I understood, historic data sets are mostly used for the runoff coefficient model? While reading I also was not sure what data is from Tuel et al. 2020b and what is new in this study. I suggest to extend the scheme in Fig. 4. Also you present results from your comparison of the satellite-based data sets in the data section. Please consider moving this in the result section.

Comment 4: Study area

You use several different domains in your manuscript. What is your snow domain (Page 6 Line 162)? What the model domain and how do the Oum-Er-Rbia watershed and the seven sub-basins fit into this? Please clarify. Add more information on the sub-basins (area, elevation,...).

Comment 5: Snow model calibration

With annual cycles (Page 6 Line 161) you refer to mean average annual cycles for the time frames you state (i.e. 1995-2005 and 2000-2011)? If yes, why do you use NSE values of average annual cycles that only partly overlap? Didn't you mention that the inter-annual variability is very high? Isn't it important to use the exact same time periods? I never saw this approach before, I think. Please explain in more detail why you use average annual cycles of snow fraction here. Furthermore, please provide more information on the calibration routine (how many runs, what NSE values did you get, what optimization algorithm,...).

Comment 6: Statistical Runoff Coefficient Model

What is the motivation to use this approach here? Please explain. What are e.g. the disadvantages compared to hydrological models, which seem predestined to investigate changes in runoff. You state that your covariates only explain 30 % of the inter-annaul variability. What about the other 70 %? Later you use this two covariates to estimates future runoff. Please justify. Why should the relationships you established for the historic time frame hold under future conditions? Don't you show that climatic

changes with fundametally alter the hydrological cycle in the region? Why do you think RH is so important in this model?

Specific comments

Page 1 Line 1-8: Large parts of the abstract are introduction. Please provide more information on your model set up and results (!) here.

Page 1 Line 11-12: What results have important implications!?

Page 1 Line 16: "1000m": Please use space between unit and number. Check your manuscript.

Page 2 LIne 26: "station snow data series": Pleas rephrase. They simulate snow for one station and use the observations from there to calibrate and validate?

Page 2 LIne 27: "somewhat less sensitive": Please rephrase.

Page 2 Line 27: "Atlas" > "High Atlas"? In general, I was a bit confused by the different term 'High Atlas', "Atlas", "Middle Atlas". Is there any difference?

Page 2 Line 34-37: With only 12 years of data no proper trend analysis can be conducted, I think. It is not very surprising that no significant trends can be found. You also states this in the following that there is a strong inter-annual variability. Maybe only inform the reader here about the strong inter-annual variability and do not discuss the results of the trend analysis from Marchane et al., 2015, as they do not seam to contain relevant information.

Page 2 Line 34: "coefficient of annual variants (0.25)": What does this coefficient mean? Where does it come from?

Page 2 Line 34: "potential long-term climate trends will be difficult to detect in such short-term series": Yes, I agree. long-term climate trends can not be determined with short time series I would recommend to rephrase to something like "A sufficient length of the time series is needed..." I still think that it is not a good idea to base your line of

argumentation here on the study analyzing 12 years of data.

Page 2 Line 36: "developed" > "assessed"?

Page 2 Line 46: "anthropogenic warming" > "global warming"?

Page 2 Line 46: "quantify the sensitivity [...] to large-scale meteo": You estimate changes in runoff.

Page 2 Line 54: Refer to the map (Fig. 1) here.

Page 2 Line 54: "4 km$^3$"; Where is this number from?

Page 3 Line 62-63: Are the 'plains' below 1000? The sentence is a bit dificult to read. Please rephrase. Do the 'plains' play any role anyway? As far as I understood you only simulate the high-head watersheds (snow domain)?

Page 3 Line 62-63: Rephrase sentence. I would present this information more neutral and remove 'precipitation is spares'. Stick to the numbers: basin average 400 mm, lowland plains 250 mm and mountains 800 mm.

Page 3 Line 68: At this point you did not introduce the data yet.

Page 3 Line 71: "somewhat persistent snowpack is not uncommon": This is a quite strange formulation. Please rephrase.

Page 3 Line 72: "rapid": What does it mean here? Are melt rates higher than in other mountain regions?

Page 3 Line 76: "seven stations": Please include table with information on stations: location, elevation,..

Page 3 Line 77: "discard": How much of the data is left after this step?

Page 3 Line 78: "daily discharge measurements": Any quality check conducted?

Page 3 Line 82: Provide average elevation and area for each sub-basin.

Page 3 Line 82: "remove the contribution...": Why do you do this? Is this a common approach?

Page 3 Line 88: What is the spatial resolution of TRMM?

Page 4 Line 110: due to cloud cover?

Page 4 Line 116: "consider" > "use"?

Page 6 Line 155: I think you should introduce this modeling domain and its characteristics in the section on study area already.

Page 6 Line 158: Why 0.8? Where is this equation from? Is this your approach?

Page 6 Line 180-183: Move to discussion?

Figure 5: What time frame do you compare here? Add elevation ranges to figures (header) directly. What fraction of the simulation are do those elevation ranges cover? Do you simulate the whole watershed or only the 'snow domain' or the seven sub-basin? Add also a legend to indicate what lines represent.

Page 8 Line 213: Where can I see 'elevation gradients' in Fig. 6?

Figure 6: Add sub-basins to map. Use grayscale? for background map. Maybe also add river network? Add figure headlines, so readers does not have to scan through the caption to find out what is shown.

Page 8 Line 215: What is a narrow band? Can you quantify?

Figure 9: What additional information do we get from Fig. 9? Why do you normalize in this way?

Page 8 Line 224-226: This sentence sound complicated. You want to point at two signals, right: total precipitation is getting less and in addition solid fraction is reduced. Both results in less snow accumulation.

Page 8 Line 228: Why do you use MCM here? Can you change to mm?
Page 8 Line 231-232: Please rephrase.

Page 8 Line 236: This section on sublimation loss contains intersting information, but somehow comes out of nowhere and I have troubles to connect it to previous parts. Please add information on your analysis on sublimation losses in your method section.

Page 9 Line 253: "may not increase very significantly": Rephrase. Maybe to "remain largely unchanged"?

Page 9 Line 277-280: This part of the discussion is confusing me. It mixes up a lot: groundwater, infiltration, evaporation, runoff concentration processes - all things you do not directly investigate in your study...

Page 9 Line 284: "The impact of decreasing RH largely dominates over that of declining snow fraction": Where can I see this. Don't they have the same effect on RCs? How robust are these findings? How much uncertainty is in your runoff estimates?

Page 10 Line 303: "Final chapter"? What final chapter?

Page 10 Line 304: "Unsurprisingly" > remove

Page 10 Line 306: "substantial mitigation of emissions": Where is this? What RCP?

Page 10 Line 308: "for much of these trends": How much? Can you quantify? How much is the contribution of changes in precipitation and how much from rising temperatures?

Page 10 Line 308: "larger snow fraction leads to less runoff": More snow results in less runoff? Do you mean lower runoff coefficients? (This is not a surprise, as you also explain). Maybe also take a look at: https://www.nature.com/articles/nclimate3225

Page 10 Line 311: remove "believed". This word is more used in the context of religion, I think.

Page 10 Line 314: Where do you show that rain-on-snow events increase? At all

elevatins? Please provide more details

Page 10 Line 317-319: Where do you show this in you analysis?

Data availability: This is not sufficient. Please provide more information on where to get the different data sets you used.

Acknowledgements: You provide information on funding here. Please do so in 'Funding information'. Acknowledge here the data providers etc.

————————————————————

---

## Short Comment (SC1) · 16 Jan 2021

Dear authors,

I also enjoyed reading your manuscript (although the results are far from enjoyable), but I would like to draw your attention to two previous works, which evaluated the impact of climate projections on runoff in High Atlas catchments.

- Rochdane, S.; Reichert, B.; Messouli, M.; Babqiqi, A.; Khebiza, M.Y. Climate Change Impacts on Water Supply and Demand in Rheraya Watershed (Morocco), with Potential Adaptation Strategies. Water 2012, 4, 28-44.

- Ayt Ougougdal, H.; Yacoubi Khebiza, M.; Messouli, M.; Lachir, A. Assessment of Future Water Demand and Supply under IPCC Climate Change and Socio-Economic

[Figure]

Scenarios, Using a Combination of Models in Ourika Watershed, High Atlas, Morocco. Water 2020, 12, 1751.

Although these studies did not focus on the snow cover, they relied on a hydrological model which has a snow routine. Hence you may find them useful to reference in your manuscript.

---

## Author Comment (AC1) · 5 Mar 2021

Dear Dr. Gascoin,

Many thanks for your comments and for pointing us to these references. The second one in particular is very interesting and would deserve to be included in a revised version of the manuscript.

Best wishes,

Alexandre Tuel (corresponding author)

———————————————

---

## Author Comment (AC2) · 12 Mar 2021

[11pt]article

graphicx url [usenames,dvipsnames]xcolor

amsfonts amssymb amsmath fix-cm latexsym array,multirow,makecell anysize rotating

P[1]>p1

[Figure]

**Answers to comments by Referee #1**

**Future projections of High Atlas snowpack and runoff under climate change**

March 12, 2021

**Comment 1** *One element that needs to be discussed in the manuscript is that the runoff coefficients are not only impacted by climatic parameters but also by surface conditions. Significant changes in vegetation cover, land use or agricultural practices often have a greater impact, and this aspect is absent from the manuscript.*

**Answer**: Every modelling framework relies on its own assumptions, which translate into uncertainties. In many climate change studies, the runoff coefficient is assumed to be more dependent on climatic parameters than on surface conditions (like vegetation) for long term changes. Still, you are correct, and it deserves to be added to the discussion. We do not explain differences in average RCs across catchments, nor can we say anything about long-term variability in RCs forced by non-climatic parameters, like land use change. Our projections also tacitly assume that parameters other than climate variables will remain the same. Still, land use changes may not be critical in the case of the High Atlas since most of the area under study is uncultivated, naturally lacking tree cover and sparsely populated. It has not experienced large-scale land use changes in the last few decades. However, climate-change driven trends in vegetation cover may still affect runoff efficiency in the region. It is also unclear to what extent enhanced groundwater pumping in the Oum-Er-Rbia watershed since the 1980s may have modified the natural water balance and, indirectly, RCs in mountain catchments.
* * *
***Comment 2*** *no reference to the literature except to the previous publication of the au-*

*thors, while an important body of work exist for the Mediterranean and Morocco :*
*Driouech et al https://doi.org/10.1007/s41748- 020-00169-3*
*Drobinski et al https://doi.org/10.1007/s10113-020-01659-w*
*Cramer et al https://doi.org/10.1038/s41558-018-0299-2*
*Lionello et al https://doi.org/10.1007/s10113-018-1290-1*

*In particular, climate projections on runoff already exist for the basin of interest, see Jaw et al https://doi.org/10.1016/j.ejrh.2015.02.008 Tramblay et al http://doi.org/10.1007/s11269-017-1870-8*

**Answer**: We thank the reviewer for the references. A more thorough literature review and comparison to previous results is required. We suggest adding the following sentences to the second paragraph of the introduction:

*"Still, climate projections over Morocco – and generally the Mediterranean – agree on robust warming and drying trends under greenhouse gas forcing (Cramer et al. 2018, Lionello et al. 2018, Drobinski et al. 2020, Tuel et al. 2020c). By the end of this century, average winter temperatures in the High Atlas could be 2-4C higher, and precipitation 25-45% lower, depending on the emissions scenario (Driouech et al. 2020, Tuel et al. 2020b)."*

*"Future trends in runoff in the High Atlas under climate change have been investigated by Jaw et al. (2015) who analyzed simulations with the Variable Infiltration Capacity model forced by regional climate model output. They found a general tendency to reductions in streamflow, with a strong sensitivity to the forcing model's precipitation trends. Tramblay et al. (2018) took a simple water balance approach, equating long-term net precipitation with water availability, to estimate future changes in dam storage across North Africa. In the High Atlas, they projected a 40-to-50% decline in water availability under business-as-usual by the end of the 21st century."*

**Comment 3** *Page 2, line 38 : what is a "parametric snow module" ?*

**Answer**: The choice of the word "module" can be confusing indeed. What we meant here is that the authors chose to include snow in their model by using a simple temperature-based parametric representation of snow accumulation and melt. We would rephrase the sentence as *"Marchane et al. (2017) developed runoff projections for the Rheyara catchment, south of Marrakech and part of the Tensift watershed, by running conceptual monthly water-balance models incorporating simple temperature-based parametrizations of snow accumulation and melt."*
* * *
**Comment 4** *Section 2.2: it is not clear why the authors consider TRMM rainfall, while*

*daily precipitation data is available at 7 locations (line 75). Why also mention CHIRPS rainfall if it is not used in the study, as the author state*

**Answer**: A gridded precipitation product is required for our simulations. While it is possible to interpolate station data using a precipitation lapse-rate, the density of stations in our study area is very small, which is why we prefer to rely on TRMM. Figure 3 discusses the adequacy of this dataset, and further details can be found in Tuel et al. 2020 J Hydrology. CHIRPS data is also used in this study to discuss the robustness of the runoff coefficient model (see Table 1).

**Comment 5** *Section 2.3: The authors should justify why they rely on only one RCM,*

*when nowadays large ensemble of climate model experiments are available, such as the Euro-CORDEX (Jacob et al https://doi.org/10.1007/s10113-020-01606-9) or Med-CORDEX (Ruti et al https://doi.org/10.1175/BAMS-D-14-00176.1) initiatives. It is well established that to obtain robust projections it is necessary to consider several combinations of GCM/RCM, and the use of only one RCM strongly reduce the relevance of the work (see Fernandez et al https://doi.org/10.1007/s00382-018-4181-8). In addition, the RegCM version 3 is rather outdated (2006) since the current version is RegCM-4 (https://www.ictp.it/research/esp/models/regcm4.aspx).*

**Answer**: We should have indeed commented on this choice. Our goal here is to build on the carefully-designed regional projections developed specifically for the region by Tuel et al. (2020). The choice of forcing GCM as well as MRCM parametrisation are discussed in detail in this reference. Also, we are not using RegCM3 but MRCM, a much improved version of this RCM. MRCM is simply based on RegCM3 the same way that RegCM4 is based on RegCM3. Admittedly, we do not explore the full range of uncertainties (warming or precipitation trends, RCM configurations, etc.) and this must be mentioned in the manuscript discussion, which we propose to do as follows:
*"The limitations of our approach introduce additional uncertainties. We rely on a single regional climate model, and thus do not explore the full range of uncertainties related to model configuration and parametrizations. Still, the regional simulations we use here*

*have been specifically tailored to the area, particularly the choice of driving GCMs, and validated against a range of observations (Tuel et al. 2020). The fact that the uncertainty in snowpack projections under RCP8.5 is small could be further explored by using other GCM/RCM combinations, especially ones that lead to less warming than projected in our three-member ensemble. As to the uncertainty in precipitation trends, it is difficult to reconcile. As shown by Tuel and Eltahir (2020), the magnitude of future wet-season precipitation in Northwestern Africa is mainly determined by that of changes in Mediterranean atmospheric circulation. Spread in dynamical trends is difficult to reduce for this region. A GCM-selection approach based on storylines could be relevant to determine plausible scenarios (Shepherd 2019)."*

**Comment 6** *Page 5, line 142: the bias correction method is not detailed. What kind of approach is used beside the use of CDFt? In such a mountainous area, and since this study consider several variables in RCM simulations (temperature, precipitation, humidity...) a pixel-by-pixel and variable-by-variable bias correction with CDFt without considering the spatial correlation and inter-variable dependencies can lead to strong uncertainties. See Vrac et al https://hess.copernicus.org/articles/22/3175/2018/ It is quite surprising that the authors seems to apply a simplistic method for bias correction of RCM outputs while they develop a high-resolution physically based framework for snow simulations.*

**Answer**: The CFDt method is the base of our bias-correction approach. You are correct to point out that our pixel-by-pixel approach does not take spatial correlation and inter-variable dependencies into account, and the R2D2 method could be interesting to apply here. R2D2 allows to correct for spatial correlations in the same variable, as well as inter-variable correlations in time. For inter-variable correlations, one challenge though is that the target datasets used in the bias-correction come from different sources (TRMM, MODIS, etc.), and thus we should be very careful about relying on their correlations when correcting the data. To that end, it would be better to correct from a single observational dataset that includes all the variables we use, but obviously such a dataset does not exist. For spatial correlations, R2D2 could improve our approach. One caveat though is that since we fit the snow model based on long-term snow cover annual cycles (instead of an actual time series of observed snow cover), the role of inter-annual variability is somewhat put aside, and the simple CDFt would probably perform just as the more complex R2D2 method (since they will both yield roughly the same seasonal and long-term values). Overall, other uncertainties (and they are admittedly many) most likely dominate.
* * *
***Comment 7*** *Page 6, line 178: similar to my comment above, how do you compute catchment-averaged October-May precipitation ? with observed rainfall or TRMM ?*

**Answer**: Catchment-averaged precipitation is computed with MRCM output data,

biased-corrected with TRMM. Station data is only used to validate the use of the TRMM dataset.
* * *
**Comment 8** *Page 6, line 177: watershed-specific fixed effects, are the parameters fixed according to size, land use etc. ?*

**Answer**: In this simple model, watershed-specific effects are not specified other than by the model intercept. We did look at whether the value of the intercept could be related to simple watershed metrics like elevation or slope distribution, land use, etc. but found no clear relationships.
* * *
**Comment 9** *Page 6, line 180: It is not clear if these sentences are results of sensitivity analysis or results of previous works*

**Answer**: The sentences from lines 180-184 relate to results from previous works and more citations, in addition to Davenport et al. 2020, should be added in a revised version (e.g., Berghuijs et al. 2017 https://doi.org/10.1002/2017WR021593; Duan et al. 2017 https://doi.org/10.5194/hess-21-5517-2017).
* * *
**Comment 10** *Page 6, line 185: Only precipitation and temperature are bias-corrected*

*? Line 125 the author state they use 6-hourly wind speed, specific humidity, air temperature, precipitation, and downward longwave and shortwave from the RCM simulations. Later on in the text, relative humidity seems to be an important driver of change, therefore better explanations on the method used to bias correct this parameter (and others) are required*

**Answer**: This is a mistake, all the variables used in the model are bias-corrected (precipitation and temperature using satellite observations, other variables using the ERA-Interim reference run. The sentence "with temperature and precipitation bias-corrected as described previously" should simply be removed.
* * *
**Comment 11** *Page 7, line 188, I don't understand this sentence "Therefore, we use the ERA/MRCM precipitation data, bias-corrected with TRMM,"*

**Answer**: The paragraph is indeed confusing. We simply mean to recall that precipitation in the MRCM runs is bias-corrected with the TRMM data (lines 125-128). These sentences would be removed in a revised version since the information is already contained in section 2.3.
* * *
**Comment 12** *Page 8, line 213: the author mention a "statistical downscaling of the*

*MRCM output to 1km,", but nothing in the method section about this. How is it possible to downscale 12km RCM simulations to 1km, with "reference" precipitation being TRMM data at 25km spatial resolution ? there is a confusion here between "downscaling" and "bias correction".*

**Answer**: More details are given in Tuel et al. 2020 J Hydrology but your remark makes it clear that we need to be more explicit about the methods. The approach involves a mixture of bias-correction and downscaling. The snow model is run at a resolution of 1km, but the bias-correction is applied at various resolutions depending on the resolution of the target datasets. Temperature is bias-corrected based on the MODIS data at a 1km resolution. Precipitation is bias-corrected at the TRMM resolution of 0.25°. Wind and humidity data are bias-corrected at the 12km resolution of the MRCM runs. Precipitation, downward longwave and shortwave, wind and humidity are then further downscaled to the MODIS 1km resolution, using equation (1) (line 138) for humidity, but keeping the same value for precipitation, radiation and wind (i.e. no elevation correction).

We suggest moving the end of section 2.3 to a new first section in the methods and to reformulate it as follows: *6-hourly wind speed, specific humidity, air temperature, precipitation, and downward longwave and shortwave are extracted from the MRCM output over our domain. For all three GCM-driven simulations, as well as the ERA-Interim driven run (hereafter referred to as ERA/MRCM), air temperature and precipitation data are bias-corrected at the 6-hourly timescale using MODIS LST-derived air*

*temperature and TRMM precipitation at their native resolutions as respective targets, via the CDF-transform method (Michelangeli et al. 2009). Bias-corrected temperature data is thus obtained at a 1km resolution, and bias-corrected precipitation data at a 0.25° resolution. Alone among the three GCMs, the IPSL-CM5A-LR model exhibits a negative bias in wet days that we correct at each grid cell by randomly generating wet days of magnitude drawn from the corresponding distribution of wet-day precipitation in the TRMM dataset. For bias correction, reference periods for "perfect" observations are 1998-2011 for TRMM and 2000-2011 for MODIS. The corresponding periods in the simulations are the same for ERA/MRCM, and the 1992-2005 and 1994-2005 periods, respectively, for each of the GCM-driven simulations. All bias corrections are performed for the cold (November-April) and warm (May-October) seasons separately. Additionally, we use wind speed, downward long- and shortwave radiation and specific humidity from the ERA/MRCM simulation over the 1982-2005 period as reference, since no observations are available. The corresponding variables in each GCM-driven simulation are therefore bias-corrected using the ERA/MRCM data as target.*

*All bias-corrected variables at resolutions coarser than the MODIS 1km grid used for the snow model are then further downscaled to a 1km resolution. Wind, radiation and precipitation data are left unchanged, but specific humidity is downscaled based on an empirical lapse-rate $\mu$ estimated at each time step:*

$$\log(q) = \log(q_{12}) + \mu \cdot (z - z_{12}) \tag{1}$$

*where $q_{12}$ is the specific humidity in a given 12-km resolution grid cell of elevation $z_{12}$, and q the downscaled value at elevation z.*
* * *
**Comment 13** *Section 4, results I see no validation of the methods applied, prior to produce future scenarios. What about the efficiency of bias-correction ? What about the efficiency of panel regression to reproduce the inter-annual variability of runoff coefficients ? Since the authors rely on TRMM rainfall, it would be interesting to see a comparison of the model driven by either observed rainfall or TRMM to reproduce discharge dynamics*

**Answer**: The efficiency of the bias-correction approach in general is discussed in detail in Tuel et al. 2020 J Hydrology, where we also compare to station data (not used in the bias-correction). We should add a reference to it in the revised version. The quality of the bias-correction is also discussed (though indirectly in the current version) when describing the performance of the individual simulations in reproducing accurate snowpack dynamics (snow cover on Figs. 5 and 6, snow-to-precipitation fraction on Fig. 10 and sublimation on Fig. 11). The results could be detailed, for instance with Table Comment 13 which shows how the GCM-driven simulations compare in terms of input data and snow model output to the observations.
Regarding the efficiency of the panel regression method, it is already shown in Figure 12 and discussed starting at line 256. However we agree that

showing the performance of the model in reproducing river discharge for the recent (post-2000) period, using TRMM precipitation, with a cross-validation approach, would be useful. Results are shown in Fig. R1. [t]

| 2*Elevation 3*All | Ann. prec. (mm)[1] | DJFM prec. (mm)[2] | Ann. snow (mm)[3] | DJFM snow (mm)[4] | Snow frac. (%)[5] |
|---|---|---|---|---|---|
| | 419 | 194 | 95 | 70 | 23 |
| | 433 | 217 | 84 | 63 | 19 |
| | 406/460 | 205/232 | 65/120 | 50/74 | 11/33 |
| 3*≥3500m | 609 | 275 | 354 | 218 | 58 |
| | 634 | 295 | 349 | 230 | 62 |
| | 566/680 | 244/352 | 271/467 | 199/265 | 40/92 |
| 3*3000-3500m | 586 | 262 | 306 | 191 | 52 |
| | 626 | 293 | 345 | 213 | 58 |
| | 564/670 | 256/326 | 226/510 | 185/245 | 36/90 |
| 3*2500-3000m | 457 | 205 | 187 | 123 | 41 |
| | 476 | 227 | 217 | 130 | 48 |
| | 426/514 | 205/240 | 127/358 | 107/146 | 28/82 |
| 3*2000-2500m | 444 | 198 | 118 | 81 | 27 |
| | 453 | 221 | 137 | 81 | 31 |
| | 414/487 | 207/235 | 77/239 | 62/101 | 17/53 |
| 3*1500-2000m | 409 | 182 | 47 | 39 | 11 |
| | 415 | 206 | 33 | 19 | 8 |
| | 394/440 | 193/225 | 17/52 | 13/23 | 4/12 |

Snow model results and input data, averaged for the whole study area and various altitudinal

bands: (1) Annual precipitation; (2) December-to-Mach (DJFM) precipitation; (3) Annual snowfall; (4) DJFM snowfall; (5) Annual fraction of solid precipitation; (6) Annual snowmelt; (7) DJFM air temperature; (8) DJFM wind speed; (9) DJFM relative humidity; (10) DJFM average snow cover; (11) DJFM mean snow water equivalent; (12) Fraction of area with $\geq$5% snow cover in DJFM; (13) Annual sublimation. For each elevation range, the top line indicates "observed" values for the 2001-2010 period (precipitation: TRMM; temperature: MODIS; wind and RH: downscaled ERA-Interim; snow cover: MODIS; and other variables: assimilated snow model results from Tuel et al. (2020a)), the middle line shows the 3-GCM average under the historical scenario (1995-2005 only), and the bottom line shows the 3-model range.

**Comment 14** *Page 10, line 280: it should be noted that the Oum Rbia basins has several areas with karstic functioning*

**Answer**: This is an important point to add to the discussion indeed. We suggest adding the following sentence on line 278: "Kartic areas are in addition quite frequent within the Oum-Er-Rbia watershed (Akdim 2015), with important implications for infiltration, aquifer and spring regimes in our study area."

***Comment 15*** *Page 10, line 294-297: this is not a result and should be in the introduc-*

*tion. The "Source: Direction de la Recherche et de la Planification de l'Eau, Rabat" is not in the reference list. This is not a result of the present study since the data and method used to obtain this result are not presented.*

**Answer**: You are correct and this sentence should be moved to the introduction.

figure-1.pdf

**Fig. R1.** Observed and predicted annual October-May discharge (MCM) for the seven sub-catchments (2001-2010). The prediction is made using TRMM precipitation and runoff coefficients predicted by the statistical runoff coefficient model fitted on 1982-2000 data only.

---

## Author Comment (AC3) · 12 Mar 2021

[11pt]article

graphicx url [usenames,dvipsnames]xcolor

amsfonts amssymb amsmath fix-cm latexsym rotating multirow

anysize

Answers to comments by Referee #2

Future projections of High Atlas snowpack
and runoff under climate change

March 12, 2021

**Comment 1** *It needs to become more clear what are the new findings of this study that extend already existing knowledge, I think.*

**Answer**: Our study brings two important contributions to the existing literature. First, we develop detailed snowpack projections for the High Atlas under future climate

change scenarios, and second, we assess the implications of a declining snowpack and enhanced aridity on regional runoff. While the runoff trends themselves are not new (it has been well established that runoff would more than likely decline in this region in the future), our methodology and assessment of the influence of snowpack on runoff coefficients both extend current knowledge. Our approach also helps quantify the uncertainty in runoff projections linked to that in runoff efficiency. We agree this needs to be made clearer in a revised version. We hope the proposed changes will answer your concerns on this point.

**Comment 2** *Although I am not a native English speaker myself, I find that the language of the manuscript needs to be improved. I came across numerous sentences and expression that I think need improvement (see specific comments). In my opinion, a language check will improve the manuscript. Furthermore, make sure to use a space to separate the unit from the number.*

**Answer**: We will revisit the language of the manuscript in a revised version and for now hope that our proposed answers will be satisfactory.

**Comment 3** *A restructuring of the text can increase its comprehensibility. I think, it would be helpful to separate Results and Discussion. First present results with subsections on snow simulations and runoff coefficient model and then discuss results in*

*a next step (maybe again use same subsection, e.g. ?snow simulations? and ?runoff coefficient model?.*

**Answer**: It is a good suggestion which we would adopt. The discussion could also be further enriched (see answers to several of the points below).
* * *
**Comment 4** *I miss a clear line of argumentation here that then leads to your aims/goals. Please restructure and rephrase so that the literature and numbers presented are easier to understand (see also specific comments). I think the main sentence is in Page 2 Line 39-40. This is your main motivation, right?*

**Answer**: You are correct. We suggest rephrasing the introduction as follows (including other remarks, especially about the literature review):
*"The High Atlas is the major source of freshwater for the semi-arid plains of central Morocco. Much of the discharge of the Oum-Er-Rbia and Tensift, the two main rivers of central Morocco, comes from the mountainous terrain where they begin their course. In this region, precipitation essentially falls at elevations above 1000 m (Boudhar et al. 2009); below that, it is scarce and evaporation is extremely high, leading to minimal runoff. Though located in a rather warm region, the High Atlas rises up to more than 4000 m and often experiences below-freezing conditions between November and March (Boudhar et al. 2009). Consequently, snow is a major component of the regional*

*water cycle (Marchane et al. 2015, Tuel et al. 2020a). It accounts for a substantial fraction of annual runoff, up to 50% in some mountain catchments (Boudhar et al. 2009), and for most of the runoff during spring, as the wet season comes to an end. Snow cover in the High Atlas is characterized by large inter-annual variability (Marchane et al. 2015, Tuel et al. 2020a), mostly following that in wet-season precipitation, itself largely shaped by the North Atlantic Oscillation (Knippertz et al. 2003, Boudhar et al. 2009).*

*However, the High Atlas snowpack may be particularly vulnerable to climate change. Climate projections over Morocco – and generally the Mediterranean – agree on robust warming and drying trends under greenhouse gas forcing (Cramer et al. 2018, Lionello et al. 2018, Drobinski et al. 2020, Tuel et al. 2020c). By the end of this century, average winter temperatures in the High Atlas could be 2-4°C higher, and precipitation 25-60% lower, depending on the emissions scenario (Ayt Ougougdal et al. 2020, Driouech et al. 2020, Tuel et al. 2020b). These combined warming and drying trends will unavoidably lead to a snowpack decline. Yet, few studies have analyzed climate change impacts on the local snowpack and regional water availability. Lopez-Moreno et al. (2017) applied a complex physically-based snow model to observed meteorological data at one station in the Moroccan High Atlas, fitted with observed snow depth at the same location. They found that High Atlas snowpack was less sensitive to warming and drying than that in other Mediterranean-climate regions (10-15% snow water equivalent decline per degree of warming), because of colder snowpack temperatures associated with high*

*latent heat losses. Still, their results pointed to a decrease in average snow duration of 25-30% and in mean Snow Water Equivalent (SWE) of 30-55% by 2050.*

*Future trends in runoff in the High Atlas under climate change have also been investigated, notably by Jaw et al. (2015) who analyzed simulations with the Variable Infiltration Capacity model forced by regional climate model output. They found a general tendency to reductions in streamflow, with a strong sensitivity to the forcing model's precipitation trends. Tramblay et al. (2018) took a simple water balance approach, equating long-term net precipitation with water availability, to estimate future changes in dam storage across North Africa. In the High Atlas, they projected a 40-to-50% decline in water availability under business-as-usual by the end of the 21st century. Only one study tried to quantify the impact of climate change on High Atlas runoff by taking snow dynamics into account: Marchane et al. 2017 developed runoff projections for the Rheyara catchment, south of Marrakech and part of the Tensift watershed, by running conceptual monthly water-balance models incorporating a simple parametric snow module. They projected a 19 to 63% decline in surface runoff by the middle of the century, dependent on model and scenario. Coupled with population growth, such trends, if realized, will inevitably translate into growing unmet water demand, as shown by Ayt Ougougdal et al. (2020) for the Ourika watershed in the High Atlas.*

*Thus, while it is clear that the region is headed towards a pronounced decline in snow-pack and runoff, much remains to be done to quantify that decline at the catchment level and reduce uncertainties. In this study, we therefore aim to develop detailed*

*snowpack projections for the High Atlas under climate change, and to assess the implications of a declining snowpack on regional runoff. We focus on the Oum-Er-Rbia watershed, a major catchment of the High Atlas. To that end, we apply the methodology of Tuel et al. (2020a) (hereafter T20a), who modeled High Atlas snowpack by applying a simple distributed snow model forced with assimilated remotely-sensed and dynamically-downscaled data. Using satellite-observed snow cover as a baseline for the current climate, we fit and run the snow model with output data from high-resolution regional climate simulations over Morocco obtained by Tuel et al. (2020b). We then quantify the sensitivity of runoff in seven mountain catchments within the Oum-Er-Rbia watershed to large-scale meteorological and snowpack conditions, and use the results to assess the impact of warming, drying and snowpack disappearance on runoff. The paper is structured as follows. Section 2 describes the study area, the data and climate model output used in this study. Section 3 presents the snow model and panel regression framework used to model runoff response to large-scale climate conditions. Snowpack and runoff projections are presented and discussed in Section 4. Finally, major results and implications are summarized in Section 6."*

**Comment 5** *This section was a bit confusing for me. You present so many different data sets with different time frames, observed, simulated, station-based, satellite-based,... Often I did not understand what you use the individual data sets for. Please support the reader by e.g. an overview scheme and/or table that illustrates what data*

*sets you use, what time frame and what for. As far as I understood, historic data sets are mostly used for the runoff coefficient model? While reading I also was not sure what data is from Tuel et al. 2020b and what is new in this study. I suggest to extend the scheme in Fig. 4. Also you present results from your comparison of the satellite-based data sets in the data section. Please consider moving this in the result section.*

**Answer**: We can add the following Table 1 to summarise the data used in this study. Also, please see Figure R1 for an update of the scheme presented in Fig. 4. The comparison of the TRMM and CHIRPS datasets is not really a result in itself; the quality of TRMM has for instance been discussed in Tuel et al. (2020a) and Ouatiki et al. (2017). Fig. 3 and the accompanying sentences are only meant to present the dataset and briefly show their adequation with the station data.
* * *
**Comment 6** *You use several different domains in your manuscript. What is your snow domain (Page 6 Line 162)? What is the model domain and how do the Oum-Er-Rbia watershed and the seven sub-basins fit into this? Please clarify. Add more information on the sub-basins (area, elevation,...).*

**Answer**: Please see the answers to Comment 28 and Comment 31. We can also update Figure 1 to show the Oum-Er-Rbia watershed, the snow domain and the sub-catchments (see Figure R2).

| Data | Description | Availability | |
|---|---|---|---|
| Station precipitation | Daily precipitation measured at seven locations in the Oum-Er-Rbia watershed | 1980-2015 | |
| Station discharge | Daily discharge measured at seven locations in the Oum-Er-Rbia watershed | 1978-2015 | |
| TRMM TMPA 3B42 version 7 | Satellite-based 3-hourly precipitation at 0.25° resolution | 1998-present | |
| CHIRPS v2.0 | Satellite- and station-based 6-hourly precipitation at 0.05° resolution | 1981-present | |
| MODIS Land Surface Temperature L3 version 6 (MOD11A1) | Satellite-based land surface temperature at 1 km resolution | 2000-present | |
| MODIS Terra snow cover daily L3 (MOD10A1) | Satellite-based fractional snow cover at 500 m resolution | 2000-present | |
| ERA/MRCM | Regional downscaling of ERA-Interim with MRCM at 12 km resolution (from Tuel et al. (2020b)) | 1981-2011 | |
| ERA/GCM | Regional downscaling of three CMIP5 GCMs (IPSL-CM5A-LR, GFDL-ESM2M and MPI-ESM-MR) with MRCM at 12 km resolution (from Tuel et al. (2020b)) | 1976-2005 (historical) and 2071-2100 (RCP4.5 and RCP8.5) | |
| SRTM 90-meter resolution version 4.1 (STRM90) | Satellite-based elevation at 90 m resolution | N/A | |

**Table 1.** Datasets used in this study

**Comment 7** *With annual cycles (Page 6 Line 161) you refer to mean average annual cycles for the time frames you state (i.e. 1995-2005 and 2000-2011)? If yes, why do you use NSE values of average annual cycles that only partly overlap? Didn't you mention that the inter-annual variability is very high? Isn't it important to use the exact same time periods? I never saw this approach before, I think. Please explain in more detail why you use average annual cycles of snow fraction here. Furthermore, please provide more information on the calibration routine (how many runs, what NSE values did you get, what optimization algorithm,...).*

**Answer**: Yes, we mean average annual cycles for the stated time periods. We rely on annual cycles instead of snow cover times series because that would not work with the RCM experiments. Instead, we assume that a correctly fitted model would yield a reasonable snow cover annual cycle over a sufficiently long period. Since the MODIS observations we use here cover 10 years (2000-2011), we simply take the last 10 years in the RCM-driven simulations to estimate the annual cycle and fit the model. Admittedly, it is arguable whether 10 years is long enough to estimate a correct annual cycle given the high inter-annual variability in snow cover in this region. By using the longer MODIS record (updated until summer 2018), we can empirically estimate the uncertainty in annual cycle when selecting 10 years of data only (Figure R3). While there are certainly differences, our choice of 10 years still seems to yield a correct

result, especially for snowpack build-up and melt.

**Comment 8** *What is the motivation to use this approach here? Please explain. What are e.g. the disadvantages compared to hydrological models, which seem predestined to investigate changes in runoff. You state that your covariates only explain 30% of the inter-annual variability. What about the other 70%? Later you use these two covariates to estimates future runoff. Please justify. Why should the relationships you established for the historic time frame hold under future conditions? Don't you show that climatic changes with fundamentally alter the hydrological cycle in the region? Why do you think RH is so important in this model?*

**Answer**: The advantage of using such a model to investigate the sensitivity and changes in runoff is its simplicity and interpretability. It allows us for instance to highlight the influence of the snow fraction in the inter-annual variability of RCs. While hydrological models are certainly more comprehensive, they also rely on many parameters which leaves them vulnerable to overfitting and thus to even more uncertainties. In addition, they suffer from the same inconvenient as our simple model, which is that there is no guarantee that the relationships (or fitting) obtained in the current climate are directly transferable to the future climate.

Furthermore, the explanatory variables used in the regression are physical variables which we understand how mechanistically relate to runoff. The difference with a hydrological model is in the different form of mathematical relations, however the physical relationship between the variables is somewhat respected. If we were to have a hydrological model, one important test for the model would have been to reproduce the observed empirical relationships that we are using. RH is important in our model because it incorporates information on both the energy constraint (evaporative demand) and on the water availability.
* * *
**Comment 9** *Page 1 Line 1-8: Large parts of the abstract are introduction. Please provide more information on your model set up and results (!) here.*

**Answer**: Here is a proposed abstract update:
*"The High Atlas, culminating at more than 4000 m, is the water tower of Morocco. While plains receive less than 400 mm of precipitation in an average year, the mountains can get twice as much, often in the form of snow between November and March. Snowmelt thus accounts for a large fraction of the river discharge in the region, particularly during spring. In parallel, future climate change projections point towards a significant decline in precipitation and enhanced warming of temperature for the area. Here, we build on previous research results on snow and climate modeling in the High Atlas to make detailed projections of snowpack and river flow response to climate change in this region. We develop end-of-century snowpack projections using a distributed energy balance snow model based on SNOW-17 and high-resolution climate simu-*

*lations over Morocco with the MIT Regional Climate model (MRCM) under a mitigation (RCP4.5) and business-as-usual (RCP8.5) scenarios. Snowpack water content is projected to decline by up to 60% under RCP4.5 and 80% under RCP8.5 as a consequence of strong warming and drying in the region. We also implement a panel regression framework to relate runoff ratios to regional meteorological conditions in seven small sub-catchments in the High Atlas. Relative humidity and the fraction of solid-to-total precipitation are found to explain about 30% of the inter-annual variability in runoff ratios. Due to projected future atmopsheric drying and the associated decline in snow-to-precipitation ratio, a 5-30% decrease in runoff ratios and 10-60% decrease in precipitation are expected to lead to severe (20-70%) declines in river discharge. Our results have important implications for water resources planning and sustainability of agriculture in this already water-stressed region."*

**Comment 10** *Page 1 Line 11-12: What results have important implications!?*

**Answer**: Please see the corrected abstract above.

**Comment 11** *Page 1 Line 16: "1000m": Please use space between unit and number. Check your manuscript.*

**Answer**: Noted, thanks.

**Comment 12** *Page 2 Line 26: "station snow data series": Please rephrase. They simulate snow for one station and use the observations from there to calibrate and validate?*

**Answer**: You are right. Here is how we would rephrase the sentence: *"Lopez-Moreno et al. (2017) applied a complex physically-based snow model to observed meteorological data at one station in the Moroccan High Atlas, fitted with observed snow depth at the same location. They found that..."*
* * *
**Comment 13** *Page 2 LIne 27: "somewhat less sensitive": Please rephrase.*

**Answer**: Here is a suggested reformulated sentence: *"They found that High Atlas snowpack was less sensitive to warming and drying than that in other Mediterranean-climate regions (10-15% snow water equivalent decline per degree of warming), because of colder snowpack temperatures associated with high latent heat losses."*
* * *
**Comment 14** *Page 2 Line 27: "Atlas" > "High Atlas"? In general, I was a bit confused by the different term ?High Atlas?, "Atlas", "Middle Atlas". Is there any difference?*

**Answer**: Good point. We suggest sticking to "High Atlas" to avoid all confusion.
* * *
**Comment 15** *Page 2 Line 34-37: With only 12 years of data no proper trend analysis*

*can be conducted, I think. It is not very surprising that no significant trends can be found. You also state this in the following that there is a strong inter-annual variability. Maybe only inform the reader here about the strong inter-annual variability and do not discuss the results of the trend analysis from Marchane et al., 2015, as they do not seam to contain relevant information.*

**Answer**: You are correct. We suggest removing this part and mentioning inter-annual variability in the introduction's first paragraph as follows:
*"The High Atlas is the major source of freshwater for the semi-arid plains of central Morocco. Much of the discharge of the Oum-Er-Rbia and Tensift, the two main rivers of central Morocco, comes from the mountainous terrain where they begin their course. In this region, precipitation essentially falls at elevations above 1000 m (Boudhar et al. 2009); below that, it is scarce and evaporation is extremely high, leading to minimal runoff. Though located in a rather warm region, the High Atlas rises up to more than 4000m and often experiences below-freezing conditions between November and March (Boudhar et al. 2009). Consequently, snow is a major component of the regional water cycle (Marchane et al. 2015, Tuel et al. 2020a). It accounts for a substantial fraction of annual runoff, up to 50% in some mountain catchments (Boudhar et al. 2009), and for most of the runoff during spring, as the wet season comes to an end. Snow cover in the High Atlas is characterized by large inter-annual variability (Marchane et*

*al. 2015, Tuel et al. 2020a), mostly following that in wet-season precipitation, itself largely shaped by the North Atlantic Oscillation (Knippertz et al. 2003, Boudhar et al. 2009)."*
* * *
**Comment 16** *Page 2 Line 34: "coefficient of annual variants (0.25)": What does this coefficient mean? Where does it come from?*

**Answer**: The coefficient of annual variation refers to the standard deviation of wet-season precipitation divided by its mean. The figure comes from https://doi.org/10.1029/2018WR022984 which we had not cited. For simplicity, we suggest removing the reference to this coefficient in the revised version.
* * *
**Comment 17** *Page 2 Line 34: "potential long-term climate trends will be difficult to detect in such short-term series": Yes, I agree. long-term climate trends can not be determined with short time series I would recommend to rephrase to something like "A sufficient length of the time series is needed..." I still think that it is not a good idea to base your line of argumentation here on the study analyzing 12 years of data.*

**Answer**: Good point, reference removed.
* * *
**Comment 18** *Page 2 Line 36: "developed" > "assessed"?*

**Answer**: We would suggest "obtained".
* * *
**Comment 19** *Page 2 Line 46: "anthropogenic warming" > "global warming"?*

**Answer**: Good suggestion, thanks.
* * *
**Comment 20** *Page 2 Line 46: "quantify the sensitivity [...] to large-scale meteo": You estimate changes in runoff.*

**Answer**: We estimate changes in runoff based on the sensitivity of runoff to meteorological variables in the current climate.
* * *
**Comment 21** *Page 2 Line 54: Refer to the map (Fig. 1) here.*

**Answer**: Noted, thanks.
* * *
**Comment 22** *Page 2 Line 54: "4 km3"; Where is this number from?*

**Answer**: This figure comes from official data (Oum-Er-Rbia watershed agency. The associated reference is the following:

Agence du Bassin Hydraulique de l'Oum-Er-Rbia: Plan Directeur d'Amenagement Integre des Ressources en Eau du Bassin de l'Oum-Er-Rbia et des bassins cotiers atlantiques, http://www.abhoer.ma/index.cfm?gen=trueid=28, 2012.
* * *
***Comment 23*** *Page 3 Line 62-63: Are the "plains" below 1000 m? The sentence is a bit difficult to read. Please rephrase. Do the "plains" play any role anyway? As far as I understood you only simulate the high-head watersheds (snow domain)?*

**Answer**: We suggest rephrasing the sentence as follows: "*The climate of the area is rather continental, characterized by a large amplitude in the annual cycle of temperature (Knippertz et al., 2003). Minimum temperatures occur in January, when they range from mild (12C) below 1000 m to cold (-5C) above 3000 m.*"
* * *
***Comment 24*** *Page 3 Line 62-63: Rephrase sentence. I would present this information more neutral and remove ?precipitation is spares?. Stick to the numbers: basin average 400 mm, lowland plains 250 mm and mountains 800 mm.*

**Answer**: Good suggestion, here is what we suggest: *Annual precipitation in the whole*

*basin averages about 400mm, with a low of 250mm in the lowland plains, and a high of 800mm in the mountains to the south*.
* * *
**Comment 25** *Page 3 Line 68: At this point you did not introduce the data yet.*

**Answer**: Correct, we will remove the reference to the figure.
* * *
**Comment 26** *Page 3 Line 71: "somewhat persistent snowpack is not uncommon": This is a quite strange formulation. Please rephrase.*

**Answer**: We suggest rephrasing as follows: *"Snowfall is common between November and March above 1500m elevation, and it is frequent ot observe snow cover persisting for several months above 2500m".*
* * *
**Comment 27** *Page 3 Line 72: "rapid": What does it mean here? Are melt rates higher than in other mountain regions?*

**Answer**: The choice of word was poor. It would be better to keep it simple: *"Melt begins in February, and the snowpack is typically gone by the end of May."*
* * *
| Name | Lon (°E) | Lat (°N) | Elevation (m) |
|------|----------|----------|---------------|
| Dechra El Oued | -5.90 | 32.68 | 595 |
| Kenifra | -5.65 | 33.00 | 1036 |
| Mechra Eddahk El Oued | -6.52 | 32.43 | 406 |
| Ouled Sidi Driss | -7.11 | 32.32 | 320 |
| Seggat | -6.69 | 31.81 | 1150 |
| Tillouguite | -6.22 | 32.02 | 1100 |
| Tizi N'Isly | -5.77 | 32.46 | 1595 |

**Table 2.** List of precipitation stations.

**Comment 28** *Page 3 Line 76: "seven stations": Please include table with information on stations: location, elevation,..*

**Answer**: We can provide this information in a supplementary table:

**Comment 29** *Page 3 Line 77: "discard": How much of the data is left after this step?*

**Answer**: Please see the answer to your next comment.

**Comment 30** *Page 3 Line 78: "daily discharge measurements": Any quality check conducted?*

**Answer**: Good point. This concerns the precipitation data as well. Here is our proposed qualification: *For each precipitation series, we conduct basic quality checks following Durre et al. 2010. We then discard all the months for which more than 10% of the data is missing or flagged. This leaves more than 95% of the data for analysis. Daily discharge measurements are available at seven locations as well, between 1978 and 2015. Each has at most 0.5% of missing data. We implement a simple quality control following Gudmundsson et al. (2018). Days with negative discharge are flagged, as well as all consecutive periods of more than 10 days during which discharge values are equal and larger than zero. We also flag as suspect daily discharge values $Q$ such that log(Q+0.01) is more than six standard deviations away from its mean value, with mean and standard deviation computed over a 10-day period around the corresponding calendar day over the whole time series.*

**Comment 31** *Page 3 Line 82: Provide average elevation and area for each sub-basin.*

| Name | Area (km$^2$) | Mean elevation (m) |
|---|---|---|
| Tarhat | 997 | 1627 |
| Chacha | 1519 | 1460 |
| Ouchene | 2391 | 1953 |
| Tillouguite | 2488 | 2363 |
| Moulay Hassan | 1700 | 2124 |
| Segmine | 506 | 1897 |
| Tamesmate | 1303 | 2198 |

**Table 3.** Characteristics of the seven analyzed sub-basins.

**Answer**: The information can be found in Table 3.

*Comment 32* *Page 3 Line 82: "remove the contribution...": Why do you do this? Is this a common approach?*

**Answer**: The base flow (aquifer discharge) contribution is important only for the Tarhat catchment (which contains the headwaters of the Oum-Er-Rbia). Removing that base flow, which hardly varies from year to year, is important to capture the inter-annual variability linked to atmospheric variables.

**Comment 33** *Page 3 Line 88: What is the spatial resolution of TRMM?*

**Answer**: Thanks for pointing out the oversight. TRMM has a resolution of 0.25°.
* * *
**Comment 34** *Page 4 Line 110: due to cloud cover?*

**Answer**: Most missing data points are due to cloud cover. We suggest reformulating as follows to avoid confusion: *"the number of missing data points (which are mainly due to the presence of clouds)"*.
* * *
**Comment 35** *Page 4 Line 116: "consider" > "use"?*

**Answer**: Good suggestion, thanks.
* * *
**Comment 36** *Page 6 Line 155: I think you should introduce this modelling domain and*

*its characteristics in the section on study area already.*

**Answer**: Good suggestion. We suggest moving the sentence to the end of section 2.1 as follows: *"We focus specifically on the 13104 km2 domain analyzed by T20a*

[Figure]

*(Fig. 1-a), which encompasses most regions within the Oum-Er-Rbia watershed that receive significant snowfall Marchane et al. 2015. Elevation in this domain ranges from 621m to 3890m. with an average of 1882m. We use a mixture of model-, station- and satellite-based hydrometeorological data, described in the following sections."*
* * *
**Comment 37** *Page 6 Line 158: Why 0.8? Where is this equation from? Is this your approach?*

**Answer**: There was a typo, the figure is 0.85 (see https://doi.org/10.1016/j.jhydrol.2020.125657) This approach was taken by Boud-har et al. (2011) (we use their value of $k$) and discussed in Tuel et al. (2020), J Hydrology: *"0.85 is selected as maximum allowable snow cover due the quasi-absence of grid-scale (500 m) snow fractions larger than 85% in satellite observations. This likely reflects the strong small- scale variability of snow cover in the High Atlas at high altitudes (Baba et al., 2019)".* The choice of parametrization for snow cover is also discussed further in the same article.
* * *
**Comment 38** *Page 6 Line 180-183: Move to discussion?*

**Answer**: Since these are considerations based on previous work, it makes sense to

leave them here (and add references, cf. point raised by the other reviewer) to justify the selection of potential covariates.
* * *
**Comment 39** *Figure 5: What time frame do you compare here? Add elevation ranges to figures (header) directly. What fraction of the simulation do those elevation ranges cover? Do you simulate the whole watershed or only the ?snow domain? or the seven sub-basins? Add also a legend to indicate what lines represent.*

**Answer**: The snow modeling domain will be explicitly shown in the revised Figure 1. It includes the seven sub-catchments. Elevation ranges correspond to the following fractions of the modelling domain: $> 3500$m: 0.2%; 3000-3500m: 2.5%; 2500-3000m: 14%; 2000-2500m: 26% and 1500-2000m: 32%.
* * *
**Comment 40** *Page 8 Line 213: Where can I see "elevation gradients" in Fig. 6?*

**Answer**: This sentence was wrongly formulated and we suggest removing it altogether.
* * *
**Comment 41** *Figure 6: Add sub-basins to map. Use grayscale? for background map.*

*Maybe also add river network? Add figure headlines, so readers does not have to scan through the caption to find out what is shown.*

**Answer**: These are good suggestions, please see the modified figure:
* * *
**Comment 42** *Page 8 Line 215: What is a narrow band? Can you quantify?*

**Answer**: Good point, the formulation was fuzzy. Here is our suggestion: *"Still, except for elevations below 2000m, mean snow cover mostly remains close (±20%) to MODIS values."*
* * *
**Comment 43** *Figure 9: What additional information do we get from Fig. 9? Why do you normalize in this way?*

**Answer**: Figure 9 allows for a more visual representation of the distribution of snowpack water content with altitude within the basin compared to Figure 8 (which focuses on each elevation band separately). We normalise by the historical total snowpack water content to highlight the relative decline projected under the two climate scenarios.
* * *
**Comment 44** *Page 8 Line 224-226: This sentence sound complicated. You want to point at two signals, right: total precipitation is getting less and in addition solid fraction is reduced. Both results in less snow accumulation.*

**Answer**: You are correct. We could reformulate it as follows: *"These SWE projections result from declines in wet-season precipitation of 25% under RCP4.5 and 40-45% un-der RCP8.5 (Tuel et al. 2020b), combined with warming trends which severely reduce the fraction of solid precipitation in the High Atlas. This is particularly true at mid-elevations (2000-2500 m) (Fig. 10) which are very close to the zero-degree line in the current climate. Warming also favors melt during winter, thus preventing the build-up of the snowpack."*
* * *
**Comment 45** *Page 8 Line 228: Why do you use MCM here? Can you change to mm?*

**Answer**: For basin-wide SWE it can be relevant to use MCM to refer to total snowpack water equivalent. For ease of comparison we suggest adding mm as a unit to the axis of Fig. 8 panels.
* * *
**Comment 46** *Page 8 Line 231-232: Please rephrase.*

**Answer**: We suggest *"Despite the spread in temperature and precipitation projections among the three models, they all agree on the virtual disappearance of the snowpack by the end of the century under business-as-usual."*
* * *
**Comment 47** *Page 8 Line 236: This section on sublimation loss contains interesting information, but somehow comes out of nowhere and I have troubles to connect it to previous parts. Please add information on your analysis on sublimation losses in your method section.*

**Answer**: That is a good point. We can briefly add to the methods the following sentence: *"In the analysis of model results, a special focus is given to current and future sublimation fluxes. Due to the particularly arid climate of the High Atlas, sublimation losses are indeed quite significant in our study area: about 9% of all snowfall on average, and up to 30% above 3500 m (Schulz et al. 2004, Lopez-Moreno et al. 2017, T20a)."* The sublimation results and discussion would then be moved to the corresponding sections.
* * *
**Comment 48** *Page 9 Line 253: "may not increase very significantly": Rephrase. Maybe to "remain largely unchanged"?*

**Answer**: Good suggestion.
* * *
**Comment 49** *Page 9 Line 277-280: This part of the discussion is confusing me. It mixes up a lot: groundwater, infiltration, evaporation, runoff concentration processes - all things you do not directly investigate in your study...*

**Answer**: Moving these few sentences in the revised version to a separate discussion section would help give them more context.
* * *
**Comment 50** *Page 9 Line 284: "The impact of decreasing RH largely dominates over that of declining snow fraction": Where can I see this. Don't they have the same effect on RCs? How robust are these findings? How much uncertainty is in your runoff estimates?*

**Answer**: Based on the sign of the regression coefficients, RH and snow fraction have opposite effects on RCs. Lower RH leads to lower RCs. In future projections, the decline in snow fraction, though very large, is not enough to compensate the effect of the RH decrease on RCs. This is a robust feature for all catchments and simulations. For instance, under RCP8.5, average relative humidity will decline by 3-6%, which will lead to an average 30% RC decline across the seven sub-catchments, while the decline

in snow fraction increases RCs by only 10%.

Regarding the full uncertainty in runoff projections, we suggest adding the following information (Fig. R7) on worst/best-case runoff changes, for instance as an additional panel in Fig. 13, and expand on it in the discussion section:
* * *
**Comment 51** *Page 10 Line 303: "Final chapter"? What final chapter?*

**Answer**: Typo! It should be "in this study".
* * *
**Comment 52** *Page 10 Line 304: "Unsurprisingly" > remove*

**Answer**: We agree.
* * *
**Comment 53** *Page 10 Line 306: "substantial mitigation of emissions": Where is this? What RCP?*

**Answer**: We were referring to the RCP4.5 scenario. It is best to reformulate the sentence as *"Given the warming and drying trends projected by climate models for this region, we find that the High Atlas snowpack will significantly decline, even in the RCP4.5 scenario which relies on substantial mitigation of emissions."*
* * *
**Comment 54** *Page 10 Line 308: "for much of these trends": How much? Can you quantify? How much is the contribution of changes in precipitation and how much from rising temperatures?*

**Answer**: A more precise quantification could be done by running the snow model under unchanged temperatures, but at first order one can argue a precipitation decline of 40-60% will lead to a snowpack decline of (at least) roughly 50%. Still, our formulation was not very robust and we suggest replacing it in the conclusion paragraph (*"Snowpack decline is evidently connected to regional warming trends, but also affected by a projected 40-60% decrease in wet-season precipitation in Northwestern Africa."*), while adding a few sentences on the topic in the discussion.

**Comment 55** *Page 10 Line 308: "larger snow fraction leads to less runoff": More snow results in less runoff? Do you mean lower runoff coefficients? (This is not a surprise, as you also explain). Maybe also take a look at: https://www.nature.com/articles/nclimate3225*

**Answer**: You are correct, one should read lower runoff coefficients.

**Comment 56** *Page 10 Line 311: remove "believed". This word is more used in the*

*context of religion, I think.*

**Answer**: We can replace it with "thought".
* * *
**Comment 57** *Page 10 Line 314: Where do you show that rain-on-snow events increase? At all elevations? Please provide more details*

**Answer**: We do not show it in our results, neither for the current nor for the future climate. In fact it is not true in future scenarios (due to the major decline in snowfall and snow cover), but it may play a role in the current climate. It is best to remove this detail from the conclusion.
* * *
**Comment 58** *Page 10 Line 317-319: Where do you show this in you analysis?*

**Answer**: We don't and again this should be moved to the discussion.
* * *
**Comment 59** *Data availability: This is not sufficient. Please provide more information on where to get the different data sets you used.*

**Answer**: ERA-Interim reanalysis data are available from https://apps.ecmwf.int/ datasets/. TRMM data are available from https://disc.gsfc.nasa.gov/datasets/TRMM_

3B42_Daily_7/summary, and CHIRPS data from https://data.chc.ucsb.edu/products/ CHIRPS-2.0/. CMIP5 model output were downloaded from https://esgf-index1.ceda. ac.uk/projects/cmip5-ceda/. MRCM simulations used in this study are available from the corresponding author upon request.
* * *
**Comment 60** *Acknowledgements: You provide information on funding here. Please do so in "Funding information". Acknowledge here the data providers etc.*

**Answer**: There is no funding section in the template and this information will be included later in the submission process.

Figure_4.pdf

Figure_1.png

Figure_test_annual_cycles.pdf

**Fig. R3.** Basin-wide snow cover annual cycle (%) estimated with MODIS data: 2000-2010 (dashed red), and median (solid blue) and 95% range (shaded) of random sampling of 1000 10-year subsets in the 2000-2018 period.

Figure_5.pdf

**Fig. R4.** Annual cycles of snow cover (in %) in the MODIS observations (2000-2010, black), ERA-Interim simulation (2000-2010, dashed red) and three GCM-driven historical simulations (1976-2005, solid blue: median; blue shading: 3-model range), at various elevations ranges within our study area: (a) > 3500 m, (b) 3000-3500 m, (c) 2500-3000 m, (d) 2000-2500 m, (e) 1500-2000 m and (f) whole area.

Figure_6.pdf

**Fig. R5.** Mean December-to-March (DJFM) fractional snow cover (%) over the basin in (a) MODIS (2000-2010) data, and (b-d) three-GCM average under the (b) historical (1976-2005), (c) RCP4.5 (2071-2100) and (d) RCP8.5 (2071-2100) experiments.

[Figure]

Figure_8.pdf

**Fig. R6.** Annual cycles of snow water equivalent (mm, left-hand axis) and corresponding total

```
Figure_runoff.png
```

**Fig. R7.** Projected runoff change (in %) for the seven catchments in the RCP4.5 (blue) and

---

## Author Response (AR2)

**Answers to comments by Referee #2**

**Future projections of High Atlas snowpack and runoff under climate change**

November 18, 2021

Dear reviewer,

We would like to thank you again for your detailed and constructive comments. We hope our replies and modifications will satisfy your remaining concerns.

Best regards,

The authors
* * *
**Comment 1** *Line 28: Not sure if 'however' fits here? At least I had to stop when I was reading it. Maybe something like 'Previous investigation indicate...' is better?*

**Answer**: You are right; we changed the sentence accordingly: "*Previous studies indicate that the High Atlas snowpack may be particularly vulnerable to climate change.*"
* * *
**Comment 2** *Fig 1: Include overview map? Not sure whether everyone knows where the Atlas and Morocco (country borders) is. Is it possible to include the information on what the red dotted line and black lines are into the legend directly? Then the reader would not have to search for it in the figure caption.*

**Answer**: Good suggestions, thanks. Please see the updated figure.
* * *
**Comment 3** *Line 83: 'variability' of the seasonal snow cover?*

**Answer**: That's correct. We made it explicit in the revised version: "*Inter-annual variability of snow cover is substantial...*"
* * *
**Comment 4** *Line 103: 'runoff will be modeled' Maybe be a bit more specific here? For a moment, I thought (again) you actually run a hydrological model for this.*

**Answer**: To avoid this misunderstanding, we suggest simply replacing "modeled" by "analysed".
* * *
**Comment 5** *Line 158: 'MRCM output must be bias-corrected' You do this in this study or was this done by T20b already? Please specify. As T20a, T20b and this study are so interlinked, is it possible to add this information into the scheme Fig. 4?*

**Answer**: We apologise if the formulation was not clear. We do indeed bias-correct MRCM output in the present study, as detailed in the rest of the paragraph. The bias-correction step is already detailed in Figure 4 (we list the datasets used as targets for the bias-correction). We modified the beginning of section 3.2.1 as follows: "*6-hourly wind speed, specific humidity, air temperature, precipitation, and downward longwave and shortwave are extracted from the MRCM output over our domain, at the 12km MRCM resolution. We then bias-correct MRCM output when reliable observations are available, and downscale it to the MODIS 1km resolution at which we run the snow model.*"
* * *
**Comment 6** *Line 174: What about temperatures? You do not downscale temperature?*

**Answer**: We do but indirectly through the bias-correction with the 1km MODIS data (so we directly obtain 1km data with no need to explicitly downscale the data). We added the following sentence at the end of section 3.2.1 to make it explicit: "*No additional downscaling for temperature data is required since the downscaling is embedded into the bias-correction step. The target MODIS LST-derived air temperature data in the bias-correction is indeed already at the required 1km resolution.*"
* * *
**Comment 7** *Line 174: What lapse-rate do you use to downscale $\mu = ?$*

**Answer**: As in T20a we use an empirically-determined lapse-rate $\mu$ estimated at each time step from the MRCM simulations.
* * *
**Comment 8** *Line 183: SWE usually used for snow water equivalent not snow water content. Please consider changing.*

**Answer**: This was a typo – SWE indeed stands for snow water equivalent in this sentence. We changed in accordingly.
* * *
**Comment 9** *Line 190: Where is this formula coming from? Why do you use this? Why 0.85 and not 0.9?*

**Answer**: This formula comes from T20a (and was also used by Boudhar et al. 2011). The reason why we choose 0.85 is that it corresponds in practice to the maximum value of MODIS gridded snow fraction in our domain. MODIS snow cover values above that threshold are extremely rare, which probably reflects the strong small-scale variability of snow cover in the High Atlas at high altitudes (Baba et al., 2019). We added this information in the revised manuscript when introducing the equation.
* * *
**Comment 10** *Line 198: Do the parameters differ a lot between the ERA and GCM calibrations? Maybe a small table with the resulting parameters?*

**Answer**: We can have it as a supplementary table indeed. Please see revised Table A2.
* * *
**Comment 11** *Fig. 5: What time frame is shown here? Can you maybe add it to the figure caption?*

**Answer**: Good point. We added this information to the caption so that readers will not have to refer to the main text: "*Annual cycles of snow cover (in %) in the MODIS observations (2000-2011; black), ERA- Interim simulation (2000-2011; dashed red) and three GCM-driven historical simulations (1994-2005; solid blue: median; blue shading: 3-model range).*"
* * *
**Comment 12** *Fig. 6 + 7 + A1: I find it hard to see the differences here. Maybe actually calculate the differences between the maps and show it?*

**Answer**: We can add a panel to Figure 7 to show the difference. Figures 6 and A1 are already quite big, so instead of adding new panels we suggest reducing the number of colors to make it more legible (please see the revised figures).
* * *
**Comment 13** *Line 252-259: Move to discussion?*

**Answer**: Given that Figure 8 includes results as well, we would prefer to keep these sentences there.
* * *
**Comment 14** *Line 416: What do you mean by 'peak snow cover' here? Maximum snow cover extent? What figure do you refer to here? I was not sure what those values represent...*

**Answer**: That's right, we did mean "maximum snow cover extent" – so we rephrased the sentence accordingly: "*Maximum snow cover extent is projected to go from 17% of the study area down to 9% under RCP4.5 and even 4% under RCP8.5 (Fig. 9-f).*"